# Anomaly Detection by an Ensemble of Random Pairs of Hyperspheres

**Walid Durani[1], Collin Leiber[2,3], Khalid Durani[4], Claudia Plant[5,6], Christian Böhm[5]**

[1]LMU Munich, Munich Center for Machine Learning (MCML), Munich, Germany
[2]Aalto University, Espoo, Finland
[3]University of Helsinki, Helsinki, Finland
[4]University of Innsbruck, Innsbruck, Austria
[5]Faculty of Computer Science, [6]ds:UniVie, University of Vienna, Vienna, Austria
durani@dbs.ifi.lmu.de, collin.leiber@aalto.fi, khalid.durani@uibk.ac.at
claudia.plant@univie.ac.at, christian.boehm@univie.ac.at

## Abstract

Anomaly detection is a crucial task in data mining, focusing on identifying data points that deviate significantly from the main patterns in the data. This paper introduces **A**nomaly **D**etection by an **E**nsemble of **R**andom Pairs of **H**yperspheres (ADERH), a new isolation-based technique leveraging two key observations: (i) anomalies are comparatively rare, and (ii) they typically deviate stronger from general patterns than normal data points. Drawing on a $\delta$-separation argument, ADERH constructs an ensemble of multi-scale hyperspheres built upon randomly paired data points to identify anomalies. To address inevitable overlaps between anomalous and normal regions in the feature space, ADERH integrates two complementary concepts: $\mathrm{Pitch}$, which highlights points near hypersphere boundaries, and $\mathrm{NDensity}$, which down-weights hyperspheres centered on sparse (and often anomalous) regions. By averaging these local, density-adjusted "isolation" indicators across many random subsets, ADERH yields robust anomaly scores that clearly separate normal from abnormal samples. Extensive experiments on diverse real-world datasets show that ADERH consistently outperforms state-of-the-art methods while maintaining linear runtime scalability and stable performance across varying hyperparameter settings.

## 1 Introduction

Anomaly detection is an essential tool in data mining, as it can uncover critical information [Agrawal and Agrawal, 2015]. For instance, anomalies may indicate credit card fraud, analyze critical behavior in network applications, or assist in diagnosing rare medical conditions [John and Naaz, 2019, Tao et al., 2018, Abuzaid, 2020]. In the era of big data, where data volumes are rapidly increasing, it is essential for anomaly detection methods to effectively and efficiently identify anomalies in large datasets [Ahmed et al., 2017, Mansour et al., 2023, Thudumu et al., 2020]. However, not all anomaly detection methods are suitable for larger datasets. Due to high runtime complexity, density-based methods like the *Local Outlier Factor* (LOF) [Breunig et al., 2000] and deep learning-based approaches [Pang et al., 2021] have limited applicability. Isolation-based methods on the other hand typically operate on data subsets, making them more effective for handling larger datasets [Xu et al., 2017, Xiong et al., 2022]. Examples of such algorithms include *Isolation Forest* (IForest) [Liu et al., 2008], *Efficient Anomaly Detection by Isolation Using Nearest Neighbour Ensemble* (INNE) [Bandaragoda et al., 2014], Extended Isolation Forest (EIF) [Hariri et al., 2019], or Deep Isolation Forest (DIF) [Xu et al., 2023a]. These approaches utilize two essential properties that distinguish anomalies from regular data points.

39th Conference on Neural Information Processing Systems (NeurIPS 2025).

- **RARITY:** Anomalies comprise only a small proportion of the dataset, i.e., most of the samples represent regular data points [Barnett et al., 1994, Aggarwal, 2016].
- **DEVIATION:** Anomalies differ significantly from the general patterns in a dataset, suggesting that they originate from different processes than regular samples [Hawkins, 1980].

We formalize these properties with the $\delta$-separation assumption: normal samples form compact regions, while anomalies lie mainly beyond their boundaries (Section 3.1). However, current isolation-based methods have certain limitations. IForest [Liu et al., 2008] efficiently detects anomalies via random partitioning, but its reliance on global, axis-aligned splits can miss complex or locally defined outliers. INNE [Bandaragoda et al., 2014] attempts to address this by utilizing hyperspheres to capture local patterns, but it is sensitive to the sample size and assigns equal weights to hyperspheres, which can limit its robustness [Bandaragoda et al., 2018].

We propose ADERH, a method that isolates anomalies using *compact hyperspheres* designed to minimize overlap with anomalies. Guided by the $\delta$-separation principle—which assumes that anomalies lie beyond normal regions —ADERH constructs small local subsets and pairs of points. By halving each pairwise distance, it forms compact hyperspheres that adapt to multiple scales and collectively cover diverse normal regions, thereby reducing overlap with anomalies and enhancing isolation precision. Since perfect $\delta$-separation may fail in practice, we refine each hypersphere's isolation signal with (i) $\mathrm{Pitch}$, a ratio-based distance measure accentuating boundary anomalies, and (ii) $\mathrm{NDensity}$, which down-weights hyperspheres in sparse (anomalous) regions. Finally, ADERH ensemble-averages these local isolation signals, further reducing variance and enhancing robustness on real-world, heterogeneous data.

**In summary, we make the following contributions:**

- We present ADERH, a novel technique for assigning anomaly scores to data points by analyzing their position within multiple hyperspheres and the characteristics of these hyperspheres.
- Hyperspheres may still include anomalies near the boundary or span around anomalies, blurring distinctions between normal and abnormal data. To overcome this, ADERH introduces two components: $\mathrm{NDensity}$, which down-weights hyperspheres in sparse (anomalous) regions, and $\mathrm{Pitch}$, which emphasizes points near hypersphere boundaries.
- Thus, ADERH more effectively distinguishes anomalies from normal samples, overcoming limitations that arise from relying solely on hypersphere- or distance-based methods.
- ADERH delivers robust and stable anomaly scores across a wide range of hyperparameters, maintains high efficiency on large-scale datasets, and—through extensive experiments involving both default parameter settings and exhaustive grid searches— outperforms state-of-the-art anomaly detection methods.

## 2 Related work

Over the past few decades, anomaly detection has been extensively studied using various techniques such as density, isolation, or deep learning.

**Isolation-based approaches** assume that a small fraction of the data consists of anomalies (**RARITY**) and that those have different attribute values than normal data points (**DEVIATION**). A prominent example is the *Isolation Forest* (IForest) [Liu et al., 2008], which recursively partitions the feature space by selecting random features and random split values; anomalies tend to have shorter paths from the root node. The *Extended Isolation Forest* (EIF) [Hariri et al., 2019] improves on IForest by using hyperplanes with randomly determined slopes for splitting, enhancing accuracy across diverse datasets. PIDForest [Gopalan et al., 2019] accelerates isolation while incorporating a density-based criterion (PIDScore) that quantifies the minimum density among all subcubes covering a data point. *Deep Isolation Forest* (DIF) [Xu et al., 2023a] leverages a learned ensemble of random representations that produce non-linear partitions in the feature space.

**Distance/Density-based approaches** flag anomalies in sparse regions. The Local Outlier Factor (LOF) [Breunig et al., 2000] measures how much a point's local density deviates from that of its neighbors, while the Connectivity-based Outlier Factor (COF) [Tang et al., 2002] refines LOF by incorporating chaining distances to better handle linear data structures.

**Boundary-based approaches** define a boundary around normal data and classify points outside this region as anomalies. For instance, the *One-Class Support Vector Machine* (OCSVM) [Schölkopf et al., 2001, Bounsiar and Madden, 2014], finds a hyperplane that maximally separates normal samples from the origin, treating any observation lying outside this boundary as anomalous.

**Ensemble-based approaches** combine multiple anomaly detection methods to mitigate individual drawbacks and leverage their strengths, to enhance performance and robustness [Zimek et al., 2014, Cheng et al., 2019, Zhao et al., 2019a]. For instance, *LODA* [Pevnỳ, 2016] aggregates outputs from diverse weak detectors, using their collective decisions to identify anomalies. Similarly, *LSCP* [Zhao et al., 2019a] selects base detectors and determines a point's anomaly score by analyzing its local data distribution and combining the detectors' outputs.

**Deep learning-based approaches** have advanced rapidly, leveraging representation learning to compute anomaly scores on complex data [Wang et al., 2019a, Pang et al., 2021]. For example, *DeepSVDD* [Ruff et al., 2018] tries to embed data into a hypersphere, classifying points on the outside as anomalies. To prevent representation collapse, *Deep Robust One-Class Classification* (DROCC) [Goyal et al., 2020] refines boundaries around normal samples by clustering them closer and using adversarial perturbations as hard negatives. Other methods emphasize collaboration or distance-based representations: *A Deep Collaborative Autoencoder Approach for Anomaly Detection* (RCA) [Liu et al., 2021] iteratively trains multiple autoencoders on low-error samples, exchanging these to enhance detection; *Unsupervised Representation Learning by Predicting Random Distances* (RDP) [Wang et al., 2019b] uses a two-branch, weight-shared model to map data into a distance-preserving space for isolating anomalies. *SLAD* [Xu et al., 2023b] introduces a self-supervised "scale" concept for tabular data, learning global normal patterns and identifying anomalies via higher errors. *Diffusion Modeling for Anomaly Detection* (DTE) [Livernoche et al., 2024] estimates how "diffused" an input is relative to the normal data manifold, enabling fast and accurate anomaly detection. UniCAD [Fang et al., 2025] introduces a unified probabilistic mixture model linking representation learning, clustering, and anomaly detection through an anomaly-aware likelihood function, yielding a theoretically grounded anomaly score.

**Hypersphere-based anomaly detection** was first introduced through global hypersphere models designed to enclose normal data points [Kumar et al., 2003, Tax and Duin, 2004]. MV-ERM and MV-SRM [Scott and Nowak, 2005] reframe minimum-volume estimation as empirical risk minimization. MV-ERM minimizes the region capturing an $\alpha$-fraction of data under a penalized risk, while MV-SRM integrates the penalty into the objective for automatic complexity control. GEM [Hero, 2006] formulated anomaly detection via geometric-entropy minimization, identifying subsets with minimal $k$-NN or MST wiring length as minimum-entropy approximations. DTM [Gu et al., 2019] estimated local radii enclosing mass $m$, with bagging reducing variance but retaining global-distance dependence. INNE [Bandaragoda et al., 2014] used hypersphere ratios—between a point's enclosing radius and its nearest neighbor's—to score anomalies. Despite progress, most methods rely on a few large hyperspheres with limited local adaptivity, often failing near or on boundaries. **In contrast**, ADERH forms an **ensemble of compact hyperspheres** from random pairs of points, each defining two half-radius spheres with varying radii. It integrates $\mathrm{Pitch}$ (boundary sensitivity) and $\mathrm{NDensity}$ (sparse-region down-weighting) to handle boundary and center anomalies. These choices ensure **robust scalability**—small subsets and simple distance checks suffice—and **strong empirical performance**, surpassing traditional and deep hypersphere methods with linear-time efficiency.

# 3 Anomaly Detection by an Ensemble of Random Pairs of Hyperspheres

Considering a dataset $\mathcal{D} \subset \mathbb{R}^d$ containing $m$ points drawn i.i.d. from a mixture distribution

$$P = \alpha P_{\mathcal{N}} + (1 - \alpha) P_{\mathcal{A}}, \quad (0 \ll \alpha < 1), \tag{1}$$

where $P_{\mathcal{N}}$ captures the *normal* data and $P_{\mathcal{A}}$ represents the *anomalous* data. Our goal is to define an *anomaly scoring function* $\mathcal{I} : \mathbb{R}^d \to \mathbb{R}$ that assigns higher scores $\mathcal{I}(x)$ to anomalous points than to normal points. Building upon the fact that anomalies form a small fraction of the data and exhibit distinctly different characteristics, we formalize these observations using a $\delta$-separation assumption.

## 3.1 $\delta$-Separation

Concretely, we assume that normal points cluster within small-radius neighborhoods, whereas anomalous points are located at least a distance $\delta$ from any local cluster:

**Assumption 3.1** ($\delta$-Separation). Let $J$ be a finite, nonempty index set and let $\{\mu_j\}_{j \in J} \subset \mathbb{R}^d$ be a set of cluster centers. Let $P_\mathcal{N}$ and $P_\mathcal{A}$ be probability measures on $\mathbb{R}^d$. Assume there exist radii $0 < \sigma < \delta$ and small parameters $0 < \varepsilon, \varepsilon' \ll 1$ such that:

1. **Normal-Point Proximity.** For $x \sim P_\mathcal{N}$,

$$\Pr\left( \min_{j \in J} \|x - \mu_j\| \leq \sigma \right) \geq 1 - \varepsilon.$$

2. **Anomaly Exclusion.** For $z \sim P_\mathcal{A}$,

$$\Pr\left( \min_{j \in J} \|z - \mu_j\| \geq \sigma + \delta \right) \geq 1 - \varepsilon'.$$

Since $\delta > \sigma > 0$, up to probabilities $\varepsilon$ and $\varepsilon'$, normal samples lie within distance $\sigma$ of some center, while anomalies lie at least $\sigma + \delta$ from every center.

**Remark:** This assumption reflects a common anomaly-detection pattern in which normal data cluster around modes $\mu_j$, and anomalies occupy sparser regions beyond distance $\delta$. For example, credit-card fraud often lies outside the compact clusters formed by legitimate transactions. Although $\delta$-separation need not hold exactly, requiring most normal points to lie within $\sigma$ of some center and most anomalies to lie beyond $\sigma + \delta$ suffices for our analysis (up to small $\varepsilon, \varepsilon'$). Similar local-separability assumptions appear in [Breunig et al., 2000, Ester et al., 1996, Bandaragoda et al., 2018]. Sections 3.3–3.4 describe how boundary- and density-based terms address partial violations of $\delta$-separation.

## 3.2 The ADERH algorithm

Building on **RARITY** and **DEVIATION** formalized by $\delta$-separation, ADERH is designed to isolate anomalies using *multiple* hyperspheres rather than a single fixed-radius sphere. Under ideal $\delta$-separation, normal points lie within radius $\sigma$, and anomalies remain at least $\delta$ away, making hyperspheres around normal samples a natural isolation mechanism. In practice, perfect separability rarely holds, so ADERH augments each hypersphere's isolation signal with $\mathrm{Pitch}$ (a ratio-based distance) and $\mathrm{NDensity}$ (a density-based term) to handle anomalies that partially overlap with normal clusters. A single hypersphere is insufficient for multi-scale data, so ADERH creates an ensemble of hyperspheres at varying radii (see Appendix B), then averages their local anomaly scores. As Theorem 3.15 shows, this ensemble averaging reduces variance and robustly isolates anomalies even when strict $\delta$-separation is violated. For this purpose, the procedure first creates a set of $n$ subsets, where each subset contains $\omega$ samples:

**Definition 3.2** (Set of subsets). We sample $n$ random subsets of size $\omega$ from the dataset $\mathcal{D}$:

$$\mathrm{SUBSETS}(\mathcal{D}, n, \omega) = \{\mathcal{S}_1, \ldots, \mathcal{S}_n\}, \tag{2}$$

where $\forall_{1 \leq i \leq n} : \mathcal{S}_i \subset \mathcal{D}, |\mathcal{S}_i| = \omega$ and $\omega$ is an even number. The subsets $\mathcal{S}_i$ are generated by uniform sampling from the dataset $\mathcal{D}$ with replacement.

The goal of ADERH is to create multiple hyperspheres of different radii in each subset $\mathcal{S}_i$, so that dense areas are captured by smaller hyperspheres, and sparser areas by larger ones. This is achieved by random pairings through the partner function $P$, which naturally reflects local density in the hypersphere radii and, therefore, yields multi-scale coverage (see Appendix B and C).

**Definition 3.3** (Partner function $P$). The function $P(x, \mathcal{S}_i)$ assigns a random point $y \in \mathcal{S}_i$ to a sample $x \in \mathcal{S}_i$, where $x \neq y$. Each partner $y$ is selected exactly once through uniform sampling. Formally, this can be expressed as $\{P(x, \mathcal{S}_i) \mid x \in \mathcal{S}_i\} = \mathcal{S}_i$. Note that $y = P(x, \mathcal{S}_i) \Rightarrow x = P(y, \mathcal{S}_i)$ is not necessarily valid.

While allowing hyperspheres to vary in radius helps capture local structures, this radius variability also risks producing oversized hyperspheres that can absorb anomalies, especially in heterogeneous

data [Bandaragoda et al., 2014, Ruff et al., 2018]. To mitigate this, ADERH transforms each pair $(x, y)$ into *two* hyperspheres—one centered on $x$ and one on $y$—each using half the pairwise distance, i.e., $\frac{1}{2} \text{dist}(x, y)$, as the radius. This halving avoids excessively large radii, reduces overall radius variance, and makes hypersphere sizes more uniform. We use the $\frac{1}{2}$ factor as a principled trade-off between coverage and exclusion, as discussed in Appendix A.

**Motivating Hypersphere Construction.** ADERH aims to isolate anomalies by combining multiple hyperspheres with diverse radii. By combining these compact hyperspheres into an ensemble, we reduce their overlap with anomalies, thereby boosting anomaly-detection performance. Below, we formalize the construction of these hyperspheres, which collectively underpin our method's ability to separate anomalies from normal data.

**Definition 3.4** (Hypersphere $\mathcal{H}$). Given a sample $x \in \mathcal{S}_i$ and its partner $y = P(x, \mathcal{S}_i)$, we create two distinct hyperspheres $\mathcal{H}(x, \mathcal{S}_i)$ and $\mathcal{H}(y, \mathcal{S}_i)$, where $x$ and $y$ are the respective centers. The radius R of both hyperspheres is defined as:

$$\text{R}(\mathcal{H}) = \frac{\text{dist}(x, y)}{2}. \tag{3}$$

Further, we define the set of potential data points that a hypersphere $\mathcal{H}$ covers as:

$$X_{\mathcal{H}} = \{x | x \in \mathbb{R}^d \wedge \text{dist}(x, \text{C}(\mathcal{H})) \leq \text{R}(\mathcal{H})\}, \tag{4}$$

where the function $\text{C}(\mathcal{H})$ returns the center of the hypersphere $\mathcal{H}$. In this paper, we employ the Euclidean distance as the distance function $\text{dist}(\cdot, \cdot)$. The ADERH algorithm, however, remains valid for any metric space, as it relies solely on the fundamental properties of a metric.

For each subset $\mathcal{S}_i$, we create an ensemble of hyperspheres:

**Definition 3.5** (Ensemble of hyperspheres $\mathcal{E}$). Let $\mathcal{S}_i \in \text{SUBSETS}$, then the ensemble of hyperspheres $\mathcal{E}$ is:

$$\mathcal{E}(\mathcal{S}_i) = \bigcup_{x \in \mathcal{S}_i} \{\mathcal{H}(x, \mathcal{S}_i), \mathcal{H}(P(x, \mathcal{S}_i), \mathcal{S}_i)\}. \tag{5}$$

The definitions set so far could be sufficient in an ideal world, where all hyperspheres were created around normal data points and the hyperspheres are sufficiently small to exclude anomalies. However, real-world datasets often violate strict $\delta$-separation. Two key complications arise:

1. **Anomaly Contamination:** Some anomalies may fall inside hyperspheres centered on normal samples (Fig. 1). Although these anomalies are technically covered, they typically appear near the hypersphere boundary rather than close to its center.

2. **Anomaly Hyperspheres:** Anomalies can also act as hypersphere centers, forming low-density (or 'sparse') hyperspheres that cover few neighbors (Fig. 2).

To handle these cases, we extend our hypersphere framework with two complementary measures Pitch (Section 3.3) and NDensity (Section 3.4).

### 3.3 Anomaly Contamination

Although $\delta$-separation outlines a margin between normal and anomalous points, real data often violates this idealized boundary [Ruff et al., 2018, Breunig et al., 2000], allowing anomalies to appear near or within local clusters (Fig. 1). Yet, on average, anomalies remain farther from hypersphere centers than normal points [Ruff et al., 2018]. To leverage this property, we use a ratio—distance from the hypersphere center over its radius—to distinguish anomalies from inliers. Concretely, for a hypersphere $\mathcal{H}$ created from subset $\mathcal{S}_i \in \text{SUBSETS}$, we define Pitch as:

**Definition 3.6** (Pitch). The Pitch represents the adjusted distance between a sample $x \in \mathcal{D}$ and the center $c = \text{C}(\mathcal{H})$ of a hypersphere $\mathcal{H}$.

$$\text{Pitch}(x, \mathcal{H}) = \begin{cases} \frac{\text{dist}(x,c)}{\text{R}(\mathcal{H})}, & \text{if } \text{dist}(x, c) \leq \text{R}(\mathcal{H}), \\ 1, & \text{otherwise.} \end{cases} \tag{6}$$

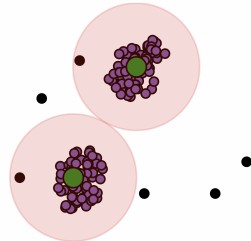

Figure 1: The issue of **Anomaly Contamination**: Anomalies (black dots) can lie within hyperspheres centered on regular samples (green dots). However, as local anomalies are typically farther from regular samples than regular samples are from each other (**DEVIATION**), we apply $\mathrm{Pitch}$ to replace strict $\delta$-separation with a ratio-based isolation measure. This flags borderline anomalies near regular samples without requiring rigid margins.

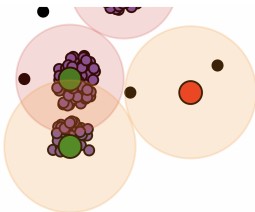

Figure 2: The issue of **Anomaly Hyperspheres**: Hyperspheres typically form around regular samples (green dots), as anomalies are rare (**RARITY**). However, in some cases, anomalies (red dots) may generate hyperspheres. Note that hyperspheres around regular samples generally enclose more points, aligning with **DEVIATION**. The colors of the hypersphere indicate which points were paired to create the hypersphere.

As anomalies are characterized by larger distances to the center of a hypersphere than normal data points, the $\mathrm{Pitch}$ strengthens the differences between corresponding samples. Abnormal data points within the hypersphere should have a large ratio $\frac{\mathrm{dist}(x,c)}{\mathrm{R}(\mathcal{H})}$ and therefore a $\mathrm{Pitch}$ close to 1. In contrast, normal data points usually have a significantly smaller $\mathrm{Pitch}$.

### 3.4 Anomaly Hyperspheres

The strategies proposed thus do not fully resolve the problem of hyperspheres centered around anomalous samples (Fig. 2). Since anomalies may exist within any random subset $\mathcal{S}_i \in \mathrm{SUBSETS}$, hyperspheres defined by these anomalies may lead to inaccurate anomaly scores, often misclassifying anomalies as normal. While the ensemble averaging across subsets mitigates their overall impact, individual subsets remain susceptible to these anomaly-centered effects. This issue is exacerbated in larger subsets, where the probability of including at least one anomaly increases with subset size $\omega = |\mathcal{S}_i|$, as shown in:

$$\mathcal{P}_{anomaly}(x) \approx 1 - \left( \frac{|\mathcal{N}|}{|\mathcal{D}|} \right)^{\omega}, \tag{7}$$

where $|\mathcal{N}|$ and $|\mathcal{D}|$ denote the number of normal samples and the dataset size, respectively. The $\mathrm{Pitch}$ determined by hyperspheres defined by these anomalies would give inaccurate anomaly scores. Considering **DEVIATION**, we know that most data points are regular samples close to each other. Anomalies are characterized by an environment of lesser density and are further away from the remaining data. Thus, the data distribution within a hypersphere indicates hyperspheres centered around anomalies. For this purpose, ADERH introduces the concept of hypersphere density:

**Definition 3.7** (Density of a hypersphere)**.** The *density* of a hypersphere $\mathcal{H}$ with associated region $X_{\mathcal{H}}$ (see Eq. 4) is defined as:

$$\mathrm{Density}(\mathcal{H}) = \frac{|X_{\mathcal{H}} \cap \mathcal{D}|}{\mathrm{R}(\mathcal{H})}. \tag{8}$$

As explained above, the greater the density of a hypersphere, the more likely it is to be a hypersphere centered around a normal sample. In contrast, a hypersphere built around an abnormal sample has a lower density. We normalize the densities by considering the density of the hypersphere with the highest density in $\mathcal{E}(\mathcal{S}_i)$. Consequently, the maximum normalized density of a hypersphere is $1$.

**Definition 3.8** (NDensity of a hypersphere)**.** The normalized density of a hypersphere is defined as:

$$\text{NDensity}(\mathcal{H}, \mathcal{S}_i) = \frac{\text{Density}(\mathcal{H})}{\max_{\mathcal{H}_j \in \mathcal{E}(\mathcal{S}_i)} \text{Density}(\mathcal{H}_j)} \tag{9}$$

## 3.5 Anomaly score

In the subsequent section, we detail the computation of the anomaly score. This score is based on the ideas of Density and Pitch, addressing both **Anomaly Contamination** and **Anomaly Hyperspheres**. First, we define the weighted Pitch (WPitch), which combines the Pitch of a sample with the NDensity of a corresponding hypersphere.

**Definition 3.9** (Weighted Pitch WPitch)**.** The weighted Pitch, denoted as WPitch, of a sample $x \in \mathcal{D}$ concerning a hypersphere $\mathcal{H} \in \mathcal{E}(\mathcal{S}_i)$ is defined as:

$$\text{WPitch}(x, \mathcal{H}, \mathcal{S}_i) = \begin{cases} (1 - \text{NDensity}(\mathcal{H}, \mathcal{S}_i)), & \text{if } x = \text{C}(\mathcal{H}), \\ (1 - \text{NDensity}(\mathcal{H}, \mathcal{S}_i)) \cdot \text{Pitch}(x, \mathcal{H}), & \text{if } x \in X_{\mathcal{H}}, \\ 1, & \text{otherwise.} \end{cases} \tag{10}$$

where $X_{\mathcal{H}}$ are the data points within $\mathcal{H}$ (Definition 3.4). Note that, since NDensity and Pitch are within the range $[0, 1]$, WPitch is also constrained to the interval $[0, 1]$.

**Weighted Pitch.** Our anomaly scoring combines a *ratio-based boundary measure* (Pitch) with a *normalized density* (NDensity) to highlight points that are both near a hypersphere boundary and in a sparse region. By adopting a multiplicative approach, high anomaly scores only occur when both boundary proximity (Pitch $\approx 1$) and hypersphere sparsity ($1 - \text{NDensity} \approx 1$) coincide, reducing the risk of overestimating anomalies in dense areas. In contrast, an additive scheme may inflate scores whenever either signal is large. As shown in Appendix R, the multiplicative form consistently achieves stronger precision and recall.

**Lemma 3.10** (Sparse anomaly–centered hyperspheres)**.** *Let $z \sim P_{\mathcal{A}}$ be an anomaly with $\min_j \|z - \mu_j\| \geq \sigma + \delta$. For a random subset $\mathcal{S}_i \subseteq \mathcal{D}$ the following holds*

$$\text{NDensity}\big(\mathcal{H}(z), \mathcal{S}_i\big) \longrightarrow 0.$$

Proof in Appendix D. Therefore, data points inside hyperspheres centered around anomalies are assigned a higher WPitch compared to data points inside hyperspheres centered around regular samples. If a point is near the center of a hypersphere with high density, the weighted Pitch (WPitch) for that point will be low, strongly suggesting it is a regular sample. Since a data point $x$ can be covered by multiple hyperspheres, it is necessary to identify the most relevant hypersphere for $x$, i.e., the one where $x$ has the minimum WPitch. This leads to the definition of the set of hyperspheres containing $x$:

$$T(x, \mathcal{S}_i) = \{\mathcal{H} \mid \mathcal{H} \in \mathcal{E}(\mathcal{S}_i) \wedge x \in X_{\mathcal{H}}\}. \tag{11}$$

From this, we compute the smallest cover (SC) as follows:

**Definition 3.11** (Smallest Cover SC)**.** We define the most relevant hypersphere for a sample $x \in \mathcal{D}$ in $\mathcal{E}(\mathcal{S}_i)$ as the smallest cover (SC), determined by:

$$\text{SC}(x, \mathcal{S}_i) = \begin{cases} \text{argmin}_{\mathcal{H} \in T(x, \mathcal{S}_i)} \text{WPitch}(x, \mathcal{H}, \mathcal{S}_i), & \text{if } T(x, \mathcal{S}_i) \neq \emptyset, \\ \emptyset, & \text{otherwise.} \end{cases} \tag{12}$$

Based on $\text{SC}(x, \mathcal{S}_i)$, we define the base anomaly score $\mathcal{F}(x, \mathcal{S}_i)$, which quantifies the likelihood of a data point being an anomaly. This score incorporates the position of $x$ within the hypersphere and the hypersphere's density.

**Definition 3.12** (Base anomaly score $\mathcal{F}(x, \mathcal{S}_i)$)**.** The base anomaly score of a data point $x$ with respect to $\mathcal{S}_i$ is denoted by $\mathcal{F}(x, \mathcal{S}_i)$ and is defined as:

$$\mathcal{F}(x, \mathcal{S}_i) = \begin{cases} \mathrm{WPitch}(x, \mathrm{SC}(x, \mathcal{S}_i), \mathcal{S}_i), & \text{if } \mathrm{SC}(x, \mathcal{S}_i) \neq \emptyset, \\ 1, & \text{otherwise.} \end{cases}$$

The base anomaly score $\mathcal{F}(x, \mathcal{S}_i)$ for a data point is bounded between $0$ and $1$.

**Lemma 3.13** (Normal and anomaly base scores)**.** *Let $x$ be a typical normal point, i.e., $x$ lies within distance $\sigma$ of some cluster center $\mu$; let $z$ be a typical anomaly, i.e., $z$ is at least $\sigma + \delta$ from every center. Suppose we draw a random subset $\mathcal{S} \subseteq \mathcal{D}$ of size $\omega$. Then, with high probability,*

$$\mathcal{F}(x, \mathcal{S}) \;<\; \mathcal{F}(z, \mathcal{S}),$$

*meaning $x$ gets a significantly lower* base anomaly score *than $z$.*

Proof in Appendix F. A single hypersphere often proves inadequate for anomaly detection in high-dimensional or heterogeneous data [Bandaragoda et al., 2014]: if its radius is too large, it may include borderline anomalies along with normal samples; if too small, it may miss broader structures. Moreover, representing the full data distribution with one hypersphere can lead to high-variance or biased anomaly scores. Instead, constructing an ensemble of hyperspheres provides multiple local characterizations at different scales, offering broader coverage and mitigating the shortcomings of any single hypersphere. This ensemble strategy also leverages variance reduction by averaging individual scores [Zimek et al., 2014], thereby diminishing noise and errors. Concretely, let $\mathcal{F}(x, \mathcal{S}_i)$ denote the *base anomaly score* of a point $x$ derived from hyperspheres created within subset $\mathcal{S}_i$. Since subsets focus on different localities and potentially produce hyperspheres of varied radii, these base scores are *independent but not identically distributed (i.n.i.d.)*. Based on this, we define the ensemble isolation score $\mathcal{I}$ as the average of the base anomaly scores across all subsets:

---

**Definition 3.14** (Ensemble Isolation Score $\mathcal{I}$)**.** The anomaly score of a sample $x \in \mathcal{D}$ is aggregated over all subsets $\mathcal{S}_i \in \mathrm{SUBSETS}$ as:

$$\mathcal{I}(x) = \frac{1}{n} \sum_{\mathcal{S}_i \in \mathrm{SUBSETS}} \mathcal{F}(x, \mathcal{S}_i), \tag{13}$$

where $n$ is the number of subsets. This ensemble-based approach minimizes the risk of anomaly scores being disproportionately influenced by any single hypersphere. The variance of the isolation score ($\mathcal{I}(x)$) is bounded by:

$$\mathrm{Var}(\mathcal{I}(x)) \leq \frac{1}{4n}. \tag{14}$$

Additionally, the probability of large deviations from the expected isolation score decreases exponentially with the number of used hyperspheres $n$:

$$P\left(|\mathcal{I}(x) - \mathbb{E}[\mathcal{I}(x)]| \geq \epsilon\right) \leq 2 \exp\left(-\frac{n\epsilon^2}{\frac{1}{2} + \frac{2}{3}\epsilon}\right). \tag{15}$$

---

Proof in Appendix E.

**Theorem 3.15** (Isolation Score Separates Normal and Anomalous Points)**.** *Let $x \sim P_{\mathcal{N}}$ lie within $\sigma$ of some $\mu_j$, and $z \sim P_{\mathcal{A}}$ lie at least $\sigma + \delta$ from every $\mu_j$. In each subset $\mathcal{S}_i$, define base scores $\mathcal{F}(x, \mathcal{S}_i)$ via the smallest cover. Then*

$$I(x) = \frac{1}{n} \sum_{i=1}^{n} \mathcal{F}(x, \mathcal{S}_i), \quad I(z) = \frac{1}{n} \sum_{i=1}^{n} \mathcal{F}(z, \mathcal{S}_i).$$

*There exist constants $\kappa_N < \kappa_A$ such that, with high probability,*

$$\mathbb{E}[I(x)] \approx \kappa_N \quad < \quad \kappa_A \approx \mathbb{E}[I(z)].$$

*Moreover, $\mathrm{Var}[I(\cdot)] \leq \frac{1}{4n}$ decreases as $n \to \infty$, making the separation robust.*

Table 1: This table reports AUC-ROC results using default parameters. Best and second-best values are shown in bold and underlined, respectively. The "AVG Rank" row lists the mean rank (lower is better). The last row shows Wilcoxon signed-rank test p-values ($\alpha = 0.05$); "+" indicates cases where ADERH performs significantly better.

| Dataset | ADERH | INNE | IForest | EIF | DIF | PIDForest | LOF | DeepSVDD | RCA | RDP | OCSVM | LODA | SLAD | DTE | UniCAD |
|---|---|---|---|---|---|---|---|---|---|---|---|---|---|---|---|
| Optdigits | **0.775 (1)** | 0.766 (2) | 0.704 (4) | 0.696 (5) | 0.588 (6) | 0.500 (13) | 0.540 (8) | 0.411 (15) | 0.740 (3) | 0.502 (12) | 0.525 (9) | 0.445 (14) | 0.560 (7) | 0.525 (9) | 0.507 (11) |
| Wbc | **1.000 (1)** | 0.911 (10) | **1.000 (1)** | **1.000 (1)** | 0.760 (13) | 0.986 (8) | 0.903 (11) | 0.901 (12) | 0.997 (6) | 0.958 (9) | **1.000 (1)** | 0.998 (5) | 0.718 (14) | 0.423 (15) | 0.994 (7) |
| Lymphography | **1.000 (1)** | 0.988 (7) | 0.998 (6) | **1.000 (1)** | 0.877 (13) | 0.977 (10) | **1.000 (1)** | 0.907 (12) | **1.000 (1)** | 0.984 (9) | **1.000 (1)** | 0.694 (14) | 0.952 (11) | 0.388 (15) | 0.988 (7) |
| Celeba | 0.732 (3) | 0.685 (7) | 0.695 (6) | 0.718 (4) | 0.663 (9) | 0.659 (10) | 0.432 (14) | 0.494 (13) | 0.664 (8) | 0.586 (11) | 0.699 (5) | 0.576 (12) | 0.787 (2) | 0.000 | **0.810 (1)** |
| Skin | 0.788 (2) | 0.707 (6) | 0.673 (10) | 0.701 (7) | 0.675 (9) | 0.723 (4) | 0.569 (11) | 0.473 (13) | 0.690 (8) | **0.810 (1)** | 0.485 (12) | 0.456 (14) | 0.766 (3) | 0.000 | 0.721 (5) |
| Pendigits | **0.962 (1)** | 0.931 (8) | 0.953 (2) | 0.947 (3) | 0.945 (5) | 0.919 (9) | 0.495 (13) | 0.238 (15) | 0.891 (12) | 0.905 (11) | 0.932 (7) | 0.946 (4) | 0.915 (10) | 0.494 (14) | 0.944 (6) |
| Wdbc | 0.981 (3) | 0.948 (10) | 0.980 (4) | 0.987 (2) | 0.722 (14) | 0.973 (7) | 0.974 (6) | 0.851 (12) | 0.950 (9) | 0.869 (11) | **0.988 (1)** | 0.978 (5) | 0.784 (13) | 0.434 (15) | 0.962 (8) |
| AD-Toothbrush | 0.901 (2) | 0.893 (3) | 0.877 (4) | 0.864 (6) | 0.877 (4) | 0.500 (14) | 0.710 (11) | 0.832 (8) | 0.682 (13) | 0.837 (7) | 0.736 (10) | 0.692 (12) | **0.937 (1)** | 0.483 (15) | 0.823 (9) |
| Wpbc | **0.554 (1)** | 0.525 (5) | 0.489 (11) | 0.506 (9) | 0.465 (15) | 0.519 (8) | 0.549 (2) | 0.474 (14) | 0.525 (5) | 0.505 (10) | 0.475 (12) | 0.533 (3) | 0.527 (4) | 0.475 (12) | 0.525 (5) |
| AD-Leather | **0.991 (1)** | 0.903 (10) | 0.982 (5) | 0.983 (4) | 0.985 (3) | 0.500 (15) | 0.794 (12) | 0.979 (7) | 0.905 (9) | 0.976 (8) | 0.884 (11) | 0.746 (13) | 0.987 (2) | 0.573 (14) | 0.980 (6) |
| Satimage-2 | **0.998 (1)** | 0.997 (3) | 0.992 (6) | 0.993 (5) | 0.996 (4) | 0.981 (7) | 0.446 (15) | 0.571 (13) | 0.974 (10) | 0.978 (9) | 0.971 (11) | 0.980 (8) | 0.917 (12) | 0.481 (14) | **0.998 (1)** |
| MNIST-C-Stripe | 0.986 (2) | 0.964 (8) | 0.966 (5) | 0.975 (4) | 0.965 (7) | 0.500 (13) | 0.425 (14) | 0.532 (12) | **0.988 (1)** | 0.900 (11) | 0.966 (5) | 0.980 (3) | 0.959 (9) | 0.367 (15) | 0.943 (10) |
| Shuttle | 0.987 (4) | 0.979 (8) | **0.997 (1)** | 0.994 (2) | 0.964 (10) | 0.966 (9) | 0.539 (14) | 0.563 (13) | 0.981 (7) | 0.954 (11) | 0.984 (5) | 0.743 (12) | 0.984 (5) | 0.000 | 0.988 (3) |
| Waveform | **0.768 (1)** | 0.740 (2) | 0.698 (8) | 0.720 (4) | 0.729 (3) | 0.593 (11) | 0.700 (7) | 0.552 (13) | 0.661 (9) | 0.589 (12) | 0.527 (14) | 0.632 (10) | 0.706 (6) | 0.497 (15) | 0.709 (5) |
| Cardio | **0.938 (1)** | 0.918 (4) | 0.919 (3) | 0.924 (2) | 0.909 (7) | 0.857 (10) | 0.665 (13) | 0.529 (14) | 0.891 (8) | 0.879 (9) | 0.917 (5) | 0.850 (12) | 0.852 (11) | 0.487 (15) | 0.912 (6) |
| AD-Bottle | 0.964 (2) | 0.936 (9) | 0.949 (6) | 0.945 (8) | 0.961 (4) | 0.500 (15) | 0.925 (10) | 0.911 (11) | 0.849 (13) | **0.977 (1)** | 0.876 (12) | 0.948 (7) | 0.963 (3) | 0.511 (14) | 0.954 (5) |
| Census | **0.628 (1)** | 0.477 (12) | 0.597 (5) | 0.621 (2) | 0.574 (7) | 0.522 (10) | 0.538 (8) | 0.497 (11) | 0.607 (4) | 0.609 (3) | 0.533 (9) | 0.467 (13) | 0.587 (6) | 0.000 | 0.000 |
| Wine | 0.839 (3) | 0.794 (5) | 0.745 (7) | 0.743 (8) | 0.448 (12) | 0.000 (15) | 0.898 (2) | 0.475 (11) | 0.802 (4) | 0.333 (14) | 0.488 (10) | 0.728 (9) | 0.762 (6) | 0.399 (13) | **0.930 (1)** |
| Musk | **1.000 (1)** | **1.000 (1)** | 0.998 (6) | 0.997 (7) | 0.977 (9) | **1.000 (1)** | 0.359 (15) | 0.691 (13) | 0.983 (8) | 0.706 (12) | 0.783 (11) | 0.898 (10) | 0.999 (5) | 0.412 (14) | **1.000 (1)** |
| AVG Rank | 1.68 | 6.32 | 5.26 | 4.42 | 8.11 | 9.95 | 9.84 | 12.21 | 7.26 | 9.00 | 7.95 | 9.47 | 6.84 | 14.11 | 5.84 |
| p-value | NA | 0.00192350 (+) | 0.00271786 (+) | 0.00354127 (+) | 0.00005341 (+) | 0.00192350 (+) | 0.00251627 (+) | 0.00005341 (+) | 0.00192350 (+) | 0.00072479 (+) | 0.00251627 (+) | 0.00005341 (+) | 0.00354127 (+) | 0.00005341 (+) | 0.02769850 (+) |

The values marked with † indicate that an error occurred during execution.

Proof in Appendix F. Ensemble averaging over all hyperspheres $\mathcal{S}_i \in$ SUBSETS reduces variance and smooths out errors from any single, poorly placed hypersphere. Thus, ADERH computes a final isolation score $\mathcal{I} \in [0, 1]$ for each point by averaging local anomaly scores, with anomalies typically scoring near 1 and normal samples near 0. Enlarging the ensemble (n) further lowers variance and enhances detection reliability. An ablation study in Appendix O and O.1 confirms that combining Pitch and NDensity effectively addresses partial violations of $\delta$-separation.

## 4  Experiments

### 4.1  Experimental setup

We perform a stratified $70\%/30\%$ train–test split that preserves the anomaly ratio, and normalize all features to the $[0, 1]$ range using a `MinMaxScaler` [Pedregosa et al., 2011]. Experiments are repeated on three stratified splits. For methods with intrinsic randomness, we run 5 seeds $\{0, 1, 2, 100, 1000\}$ per split (15 runs total per dataset–method), while deterministic methods use 3 runs (one per split). Models are trained on the training partition and produce continuous anomaly scores on the test partition. We report AUC-ROC and AUC-PR [Davis and Goadrich, 2006] as *mean* across runs. For the experiments, we applied default parameters following the respective publications, with ADERH's parameters detailed in Appendix H. We also conducted experiments using a comprehensive grid search (details in Appendix Q). Across datasets, ADERH was compared to all competitors using a paired Wilcoxon signed-rank test with Holm–Bonferroni correction at $\alpha$=0.05 [McDonald, 2014]. Experimental details, including runs, seeds, and significance testing, are provided in Appendix K. Code is available at `https://github.com/Walid10010/ADERH.git`.

#### 4.1.1  Real-world datasets

Tables 1 and Appendix L present the AUC-ROC and AUC-PR results under default settings. Notably, ADERH achieves first place in 11 datasets and second place in 6 for AUC-ROC (Table 1), outperforming isolation-based methods (e.g., IForest) and deep anomaly detection methods (e.g., DeepSVDD, RCA, DIF). Compared to single-sphere or single-hyperplane strategies (e.g., DeepSVDD, OCSVM), ADERH's ensemble of hyperspheres excels by incorporating each hypersphere's position and unique weight (WPitch). Unlike INNE, which relies solely on the ratio of two hyperspheres' radii, ADERH forms pairs of compact (half-radius) hyperspheres and augments their scores with Pitch and NDensity, enabling it to better highlight borderline anomalies and down-weight sparse, anomaly-centered hyperspheres, thereby producing more accurate anomaly scores. Further, while LOF relies on a fixed-size neighborhood, ADERH forms an ensemble of WPitch-weighted hyperspheres via random pairing, creating robust multi-scale coverage and yielding a top AUC-ROC rank of 1.68 (on average). ADERH also excels in AUC-PR (Appendix L), with an average rank of 2.29. Wilcoxon signed-rank tests confirm that ADERH significantly outperforms its competitors in both AUC-ROC and AUC-PR.

Table 2: AUC-ROC (higher is better) comparing ADERH against classical covering methods GEM and DTM. Numbers in parentheses are per-row ranks; ties share the same rank.

| Dataset | ADERH | GEM | DTM |
|---|---|---|---|
| Optdigits | **0.775 (1)** | 0.378 (3) | 0.770 (2) |
| Skin | **0.788 (1)** | 0.613 (3) | 0.784 (2) |
| Pendigits | **0.962 (1)** | 0.714 (3) | 0.960 (2) |
| AD-Toothbrush | 0.901 (2) | **0.919 (1)** | 0.870 (3) |
| Wpbc | **0.554 (1)** | 0.513 (3) | 0.536 (2) |
| AD-Leather | 0.991 (2) | 0.991 (2) | **0.992 (1)** |
| Satimage-2 | **0.998 (1)** | 0.895 (2) | **0.998 (1)** |
| Backdoor | **0.889 (1)** | 0.664 (3) | 0.852 (2) |
| Waveform | **0.768 (1)** | 0.708 (3) | 0.743 (2) |
| Cardio | **0.938 (1)** | 0.663 (3) | 0.927 (2) |
| AD-Bottle | **0.964 (1)** | 0.958 (3) | 0.963 (2) |
| Celeba | **0.732 (1)** | 0.570 (3) | 0.714 (2) |

**Comparison with Classical Covering Methods (GEM, DTM)**    Table 2 compares ADERH with two classical covering baselines, GEM and DTM. ADERH attains the top performance per-dataset AUC-ROC on the majority of datasets, reflecting the benefit of its *multi-scale hypersphere* coverage induced by random pairing and the multiplicative $\text{Pitch} \times \text{NDensity}$ score, which together sharpen separation between nominal and anomalous regions.

**Cross-dataset stability.**    To quantify robustness (Table 3), we summarize the *average* standard deviation across all datasets for the main competing methods below (lower is better). ADERH achieves the lowest variability on both AUC-ROC and AUC-PR.

Table 3: **Average standard deviation across datasets.** Relative to INNE and IForest, ADERH indicates greater stability and consistency.

| Method | Mean AUC-ROC std | Mean AUC-PR std |
|---|---|---|
| ADERH | **0.0133** | **0.0317** |
| INNE | 0.0241 | 0.0515 |
| IForest | 0.0235 | 0.0417 |

## 5   Conclusion

In this paper, we introduce ADERH, a novel isolation-based anomaly detection method that leverages the core characteristics of anomalies: **RARITY** and **DEVIATION**. By utilizing hyperspheres and the concepts of $\text{Pitch}$ and $\text{NDensity}$, ADERH delivers precise and reliable anomaly scores. Extensive experiments demonstrate its superiority over state-of-the-art methods across diverse datasets, consistently achieving higher AUC-ROC and AUC-PR scores than its competitors. Additionally, ADERH is robust to parameter variations and scales linearly with dataset size, making it highly practical for large, high-dimensional datasets.

## 6   Limitations

Like other distance-based anomaly detectors (e.g., Isolation Forest, LOF), ADERH is affected by the curse of dimensionality, where distance concentration weakens inlier–outlier contrast. Through its integration of multi-scale hypersphere modeling, geometry-aware scoring, and density-sensitive aggregation, ADERH achieves strong performance on high-dimensional benchmarks such as AD-Leather, AD-Toothbrush, and Census. Nevertheless, dimensionality remains a fundamental challenge; Appendix T discusses future directions toward dimensionality-aware and structure-preserving models.

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

| Appendix Section | Content |
|---|---|
| Appendix A | Balanced shrinkage of pairwise distances |
| Appendix B | Multi-Scale coverage and justification for random pairing in hypersphere ensembles |
| Appendix C | Analyzing the distribution of the radii of hyperspheres created by ADERH |
| Appendix D | Proof of Lemma 3.10 |
| Appendix E | Variance reduction and error bounds for the ensemble anomaly score $\mathcal{I}$ |
| Appendix F | Proof of Theorem 3.15 |
| Appendix G | Algorithmic details |
| Appendix H | Parameter setting |
| Appendix I | Robustness |
| Appendix J | Datasets |
| Appendix K | Experimental details |
| Appendix L | AUC-PR Results |
| Appendix M | Runtime complexity |
| Appendix N | Runtime experiments |
| Appendix O | Ablation study: different settings of ADERH |
| Appendix P | Ensembling improves anomaly detection over any single subset |
| Appendix Q | Grid search experiment for isolation and non-isolation methods |
| Appendix R | Ablation study: multiplicative vs. additive Fusion |
| Appendix S | Limitations |
| Appendix T | Future work |

Table 4: Structure of the appendix.

# A  Balanced shrinkage of pairwise distances

We present a mathematical argument indicating that among all scaling factors $\alpha \in (0, 1]$, setting $\alpha = \frac{1}{2}$ provides the best balance between (i) ensuring that hyperspheres remain compact enough to avoid covering anomalies, and (ii) retaining enough coverage to include normal points within the same local region. Our analysis assumes a typical "$\delta$-separation" setting where normal clusters have radius at most $\sigma$, and anomalies lie beyond $\sigma + \delta$ from each cluster center.

$\delta$-**assumption:** Assume each normal cluster has radius $\sigma$. That is, any two normal points $x, y$ in the same cluster satisfy

$$\|x - y\| \leq 2\sigma,$$

since each lies within distance $\sigma$ of the same center. Anomalies lie at least $\sigma + \delta$ away from every cluster center, with $\delta > 0$.

**Shrinking Factor** $\alpha \in (0, 1]$**:** Given a pair $(x, y)$ of points, suppose we define a hypersphere with radius

$$\alpha \|x - y\|.$$

We compare different values of $\alpha$ in $(0, 1]$.

**Lemma: coverage criterion for normal pairs**

**Lemma A.1.** *Let $x, y$ be two normal points from the same cluster, with $\|x - y\| \leq 2\sigma$. If $\alpha \|x - y\| \leq \sigma$, then this hypersphere fully covers the local region of radius $\sigma$ around one center. Equivalently,*

$$\alpha \leq \frac{\sigma}{\|x - y\|} \leq \frac{\sigma}{\sigma} = 1$$

*if $\|x - y\| \leq 2\sigma$. In particular, if $\alpha = \frac{1}{2}$, then*

$$\alpha \|x - y\| \leq \sigma \quad whenever \ \|x - y\| \leq 2\sigma.$$

*Sketch.* Since $\|x - y\| \leq 2\sigma$, multiplying by $\frac{1}{2}$ (or any $\alpha \leq \frac{1}{2}$) ensures the radius does not exceed $\sigma$. Hence, normal points within that cluster remain inside or near the hypersphere, supporting good coverage of normal data.

**Lemma: exclusion criterion for anomalies**

**Lemma A.2.** *Let $z$ be an anomaly with distance at least $\sigma + \delta$ from every normal cluster center, and let $x$ be a normal point in some cluster. If $\|x - z\| \geq \delta$, then any hypersphere with radius strictly below $\delta$ around $x$ will not include $z$. In particular, if $\alpha \|x - y\| \leq \delta$ for normal points $x, y$, then a distant anomaly $z$ remains outside that hypersphere.*

*Sketch.* From the typical $\delta$-separation assumption, normal–anomaly distances exceed $\delta$. Thus, if the hypersphere radius is at most $\delta$, the anomaly cannot lie inside the same hypersphere.

**Balancing coverage vs. exclusion**

We want $\alpha \|x - y\|$ to be $\leq \sigma$ for normal–normal pairs (to ensure good coverage), yet also $\leq \delta$ (or at least not too large) so that anomalies do not get unintentionally included. Setting $\alpha = \frac{1}{2}$ provides a natural boundary:

1. $\|x - y\| \leq 2\sigma \implies \frac{1}{2}\|x - y\| \leq \sigma$, so normal points in the same cluster remain covered.
2. If $\|x - y\| \approx 2(\sigma + \delta)$, halving prevents the radius from reaching $\sigma + \delta$. Therefore, a hypersphere centered on a normal point is less likely to include anomalies that lie beyond $\sigma + \delta$.

By contrast, if $\alpha < \frac{1}{2}$, we risk under-covering normal points (the radius becomes too small, potentially splitting the cluster). If $\alpha > \frac{1}{2}$, the radius can exceed $\sigma$, enlarging hyperspheres such that anomalies may sneak inside.

**Proposition A.3.** *Under the conditions of Lemmas A.1 and A.2, consider $\alpha \in (0, 1]$ as a scaling factor for the pairwise distance $\|x - y\|$. Setting $\alpha = \frac{1}{2}$ ensures both:*

1. *Adequate local coverage of normal–normal pairs, since $\alpha\|x - y\| \leq \sigma$ whenever $\|x - y\| \leq 2\sigma$,*

2. *Limited overshoot for larger distances, so that hyperspheres around normal points are less likely to include anomalies lying beyond $\sigma + \delta$.*

*Thus, $\alpha = \frac{1}{2}$ provides a balanced trade-off between cluster coverage and anomaly exclusion, though not necessarily an optimal choice in a formal sense.*

*Sketch.* For $\alpha < \frac{1}{2}$, hyperspheres become smaller than $\sigma$ even when $\|x - y\| \leq 2\sigma$, under-covering normal regions. For $\alpha > \frac{1}{2}$, hyperspheres can exceed $\sigma$, risking inclusion of anomalies. Thus $\alpha = \frac{1}{2}$ is the threshold guaranteeing cluster coverage without inflating radii enough to merge anomalies.

**Empirical evidence** Our theoretical discussion is corroborated by the experimental results in Table 5, where $\alpha = \frac{1}{2}$ consistently demonstrates strong performance. The table compares ADERH scores across four scaling factors (0.25, 0.5, 0.75, and 1.00). Despite partial violations of strict $\delta$-separation (e.g., overlapping clusters or varied cluster sizes), $\alpha = \frac{1}{2}$ attains the best average rank, consistently striking a balance between sufficiently covering normal clusters and limiting radius overshoot that includes anomalies. In practice, halving the pairwise distance still prevents hyperspheres from becoming too large and diluting their ability to isolate anomalies, reinforcing $\alpha = \frac{1}{2}$ as a robust heuristic—even beyond the perfect $\delta$-separation setting.

**Conclusion**

Mathematically, halving the distance $\|x - y\|$ avoids excessively large hyperspheres that might encompass anomalies, while still ensuring that two normal points within the same local cluster remain covered. Any fraction $\alpha < \frac{1}{2}$ sacrifices some coverage of normal data, and any $\alpha > \frac{1}{2}$ raises the risk of anomaly inclusion. Hence $\alpha = \frac{1}{2}$ emerges as a compromise for multi-scale isolation.

# B  Multi-Scale Coverage and Justification for random pairing

In this appendix, we provide a mathematically grounded rationale for using *random pairwise distances* as the basis for local structures (e.g., hyperspheres) in anomaly detection. We show how sampling

Table 5: The table shows the results for ADERH for different radius scalings (0.25, 0.5, 0.75, 1.00).

| Dataset | ADERH | ADERH-0.25 | ADERH-0.75 | ADERH-1.00 |
|---|---|---|---|---|
| Lymphography | **1.000 (1)** | **1.000 (1)** | **1.000 (1)** | 0.833 (4) |
| Pendigits | **0.309 (1)** | 0.305 (2) | 0.294 (3) | 0.291 (4) |
| AD-Toothbrush | **0.840 (1)** | 0.783 (4) | 0.826 (2) | 0.786 (3) |
| Wpbc | **0.261 (1)** | 0.258 (3) | 0.260 (2) | 0.255 (4) |
| AD-Leather | **0.975 (1)** | 0.970 (3) | 0.973 (2) | 0.954 (4) |
| Backdoor | 0.222 (2) | **0.312 (1)** | 0.221 (3) | 0.193 (4) |
| Cardio | **0.588 (1)** | 0.532 (4) | 0.586 (2) | 0.554 (3) |
| AD-Bottle | 0.940 (2) | 0.914 (4) | **0.943 (1)** | 0.928 (3) |
| Census | 0.075 (2) | **0.081 (1)** | 0.073 (3) | 0.000 |
| Musk | **1.000 (1)** | **1.000 (1)** | **1.000 (1)** | **1.000 (1)** |
| Glass | 0.222 (2) | 0.111 (4) | 0.221 (3) | **0.223 (1)** |
| AVG Rank | 1.36 | 2.55 | 2.09 | 3.18 |

pairs $(X, Y)$ at random from a dataset naturally spans a wide range of distances, thereby offering multi-scale coverage with minimal manual tuning.

### Random pairwise distances

Let $\mathcal{D} \subset \mathbb{R}^d$ be drawn from an unknown distribution $P$. Define two i.i.d. random variables $X, Y \sim P$, and consider the distance

$$D = \|X - Y\|.$$

Our goal is to approximate the distribution of $D$ by randomly pairing points in small subsets of $\mathcal{D}$. Concretely:

- **Subset Selection:** Choose a small subset $\mathcal{T} \subseteq \mathcal{D}$ of size $\omega$.
- **Random Partnering:** For each $x \in \mathcal{T}$, select a partner $y \in \mathcal{T}$ uniformly at random.
- **Distance Extraction:** Record the distances $\|x - y\|$. Repeating over multiple subsets yields a set of pairwise distances approximating $F_D$, the distribution of $\|X - Y\|$ in the entire dataset.

### Theoretical underpinnings

**Lemma B.1** (Short Distances: Intra-Cluster Pairs). *Let $C \subset \mathbb{R}^d$ be a set of points with diameter $\sigma$, meaning $\|x - y\| \leq \sigma$ for all $x, y \in C$. If $\Pr(X \in C) = \alpha > 0$, then*

$$\Pr(D \leq \sigma) \geq \alpha^2.$$

Since $X, Y$ are i.i.d., $\Pr(X \in C, \ Y \in C) = \alpha^2$. Inside $C$, all distances are $\leq \sigma$. Hence $\Pr(D \leq \sigma) \geq \alpha^2$.

**Lemma B.2** (Inter-cluster pairs). *Let $C_1, C_2 \subset \mathbb{R}^d$ be disjoint sets with*

$$\Delta = \min_{x \in C_1, \, y \in C_2} \|x - y\| > 0, \qquad \Pr(X \in C_1) = \alpha_1, \ \Pr(X \in C_2) = \alpha_2.$$

*Then*

$$\Pr(D \geq \Delta) \geq 2\,\alpha_1\alpha_2.$$

*Proof.* Because $(X, Y)$ are i.i.d.,

$$\Pr(X \in C_1, \ Y \in C_2) = \alpha_1\alpha_2, \qquad \Pr(X \in C_2, \ Y \in C_1) = \alpha_2\alpha_1.$$

The two events are disjoint and each guarantees $\|X - Y\| \geq \Delta$. Summing them yields the stated lower bound. $\square$

For any pair $(X \in C_1, Y \in C_2)$, the distance is at least $\Delta > 0$. The probability of drawing such a pair is $\alpha_1\alpha_2$. Thus $\Pr(D \geq \Delta) \geq 2\alpha_1\alpha_2$.

**Multi-Scale coverage**

**Theorem B.3.** *Combining Lemmas B.1 and B.2 shows that if the data contains multiple clusters or distinct subregions, random pairs inevitably yield both small distances (within clusters) and large distances (across clusters). Thus, the distribution of D spans a continuum from local to global scales in proportion to the dataset's mixture structure. No single global radius σ needs to be chosen a priori, as the data itself reveals numerous scales.*

**Benefits of random pairing**

The use of random pairwise distances offers several key benefits in anomaly detection. First, it provides a data-driven, multi-scale representation, as sampling pairs from the empirical distribution of $\|X - Y\|$ inherently captures both small (intra-cluster) and large (inter-cluster) distances. Consequently, one need not pre-specify a single global threshold or neighborhood size, which is especially important in the presence of heterogeneous cluster densities. In addition, random pairing naturally accommodates multiple, possibly irregularly shaped clusters, since each subset-based pairing reflects local geometric structures without requiring exhaustive distance computations or fully global operations. Repeated sampling of these pairs further promotes broad coverage of the data distribution, ensuring that relevant scales—ranging from tight local neighborhoods to more expansive separations—are collectively included in the modeling. Practically, the process is also computationally simple, as each subset only requires $\omega$ points, and pairing them is straightforward; this avoids the expense of building full distance matrices on the entire dataset. Overall, random pairing thus integrates local adaptivity, multi-scale sensitivity, and computational efficiency in a single procedure, making it both robust and scalable for real-world anomaly detection scenarios.

**Empirical confirmation**

In *Appendix C*, we illustrate this behavior using real-world data. The distribution of distances $\|X - Y\|$ is often multi-modal. Small-distance peaks align with compact clusters, while heavy tails arise from inter-cluster distances or outliers, confirming the guarantees given in Lemmas A.1–A.2.

## B.1 Concluding remarks

Random pairing of points is a simple yet powerful tool for capturing the *full range* of distances in a dataset. This mechanism organically yields small distances in dense clusters and larger ones across sparser regions. We emphasize:

- **No single scale** must be predetermined.
- **Multi-scale structure** emerges directly from the distribution of $\|X - Y\|$.
- **Cluster shapes and heterogeneity** are naturally captured—crucial for effective anomaly detection.

## C Analyzing the distribution of the radii of hyperspheres created by ADERH

Having established in Section B that random pairing of points theoretically enables multi-scale coverage, we now illustrate these claims with real-world data. Specifically, we examine the distribution of hypersphere radii generated by ADERH via randomly paired points. As shown in Figure 3, the resulting radii indeed cover a broad range, validating our theoretical analysis in three ways:

1. **Small intra-cluster radii.** In denser regions of the data, random pairs of nearby points produce hyperspheres with small radii, capturing fine-grained local neighborhoods.
2. **Large inter-cluster radii.** In regions separating distinct clusters or featuring anomalies, random pairs tend to yield comparatively larger radii, thereby modeling more global scales.
3. **Mixed scales in heterogeneous data.** In practice, many real-world datasets exhibit multiple, potentially overlapping clusters of various densities. Our empirical results indicate that even a modest number of random subsets and pairings is sufficient to uncover hyperspheres at numerous scales simultaneously.

Overall, these observations lend strong empirical support to the multi-scale coverage facilitated by our random pairing strategy. They also highlight how data-specific structure—whether it is tight clusters, more diffuse distributions, or the presence of outliers—naturally arises in the empirical distribution of pairwise distances. Hence, without requiring explicit parameter tuning for a single global scale, our procedure effectively adapts to the intrinsic geometry of each dataset.

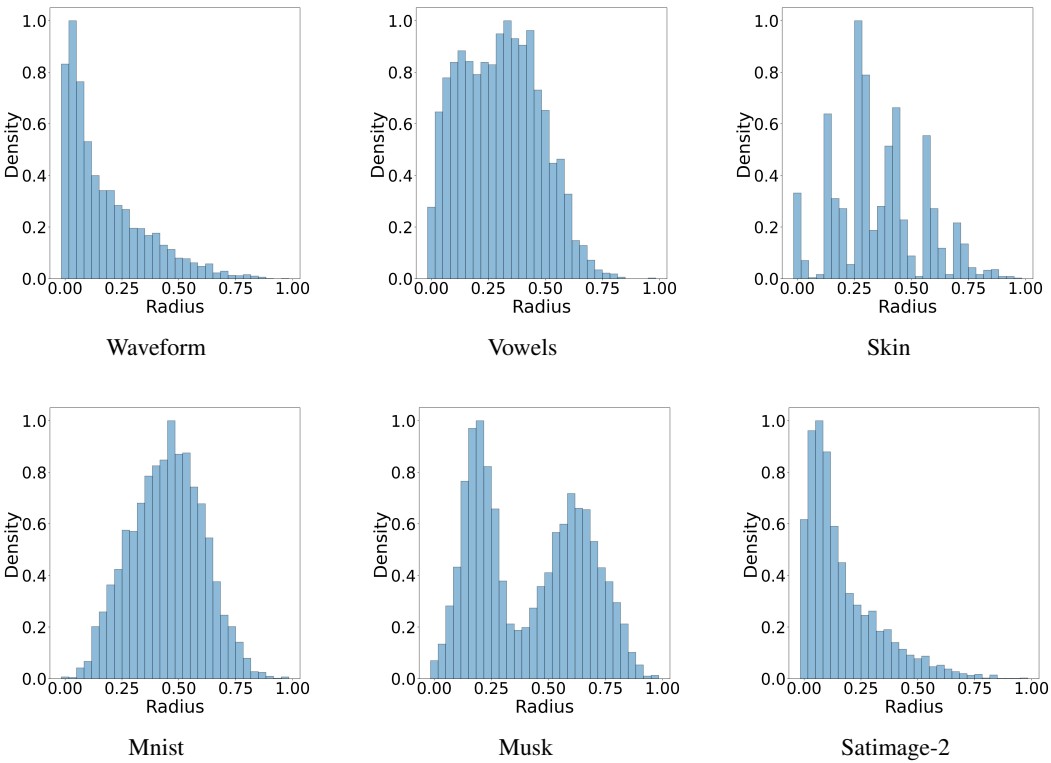

Figure 3: The distribution of the radii of hyperspheres created by ADERH for each dataset (scaled to $[0, 1]$).

# D  Proof for Lemma 3.10

*Proof.* Let $\mathcal{D}$ be the dataset, and let $\mathcal{S}_i \subseteq \mathcal{D}$ be a (random) sample of size $|\mathcal{S}_i|$.

**Separation assumption.** By Definition 3.1, any anomaly $z$ satisfies

$$\mathrm{dist}(z, \mu_j) \geq \sigma + \delta \quad \text{for every normal cluster center } \mu_j.$$

Hence, the distance from $z$ to the boundary of any $\sigma$-radius normal cluster is at least $\delta$.

**Sparse coverage by anomaly-centered hyperspheres.** Choose a radius $r$ such that

$$r = \gamma\sigma \quad \text{with} \quad \gamma < \frac{\delta}{\sigma}.$$

Since $\gamma\sigma < \delta$, any ball of radius $r$ around an anomaly $z$, i.e. the set

$$\{\, x : \|x - z\| \leq r\},$$

does *not* intersect (or barely intersects) the normal clusters. Indeed, each normal cluster of radius $\sigma$ is at least $\delta$ away from $z$, and $\delta - r > 0$. Therefore,

$$\left|\{\, x \in \mathcal{S}_i : \|x - z\| \leq r\}\right| \approx 0 \quad \text{with high probability.}$$

Formally, by Chernoff or Hoeffding bounds, the probability that a random sample of normal points places more than a negligible number of points inside $\{x : \|x - z\| \leq r\}$ **decays exponentially** in $|\mathcal{S}_i|$. Thus, the hypersphere $\mathcal{H}(z)$ of radius $r$ around $z$ captures almost no normal points, making its density

$$\mathrm{Density}\big(\mathcal{H}(z)\big) = \frac{\left|\{\, x \in \mathcal{D} : \|x - z\| \leq r\}\right|}{r} \approx 0.$$

**High coverage by normal-centered hyperspheres of similar radius.** Next, consider a normal point $y$ lying near its normal cluster center $\mu_j$ with $\|y - \mu_j\| \leq \sigma$. For $r = \gamma\sigma$ (the same radius as above), the ball

$$\{\, x : \|x - y\| \leq r\}$$

fully contains (or nearly contains) the $\sigma$-radius cluster around $\mu_j$. Consequently, there are many normal points in that ball, and so a hypersphere $\mathcal{H}'$ of radius $r$ centered on such a normal $y$ will have substantially higher density:

$$\mathrm{Density}\big(\mathcal{H}'\big) > \mathrm{Density}\big(\mathcal{H}(z)\big).$$

Let $\mathcal{E}_{\mathcal{H}}(\mathcal{S}_i)$ denote the set of all hyperspheres of radius $r$ centered at points in $\mathcal{S}_i$. Then

$$\max_{\mathcal{H}' \in \mathcal{E}_{\mathcal{H}}(\mathcal{S}_i)} \mathrm{Density}(\mathcal{H}') > \mathrm{Density}(\mathcal{H}(z)).$$

**Normalized density goes to zero.** Define the normalized density of the anomaly-centered hypersphere by

$$\mathrm{NDensity}(\mathcal{H}(z)) = \frac{\mathrm{Density}(\mathcal{H}(z))}{\max_{\mathcal{H}' \in \mathcal{E}_{\mathcal{H}}(\mathcal{S}_i)} \mathrm{Density}(\mathcal{H}')}.$$

Since the density of $\mathcal{H}(z)$ is extremely small while there exist normal-centered hyperspheres of radius $r$ with significantly higher density, we conclude that

$$\text{NDensity}(\mathcal{H}(z)) \approx 0.$$

Moreover, by standard concentration arguments (e.g. laws of large numbers for the fraction of points in a given region), this low-density phenomenon holds with high probability over the random choice of $\mathcal{S}_i$. As $|\mathcal{S}_i| \to \infty$, the probability that any anomaly-centered sphere has more than a negligible fraction of normal points converges to 0. Therefore,

$$\text{NDensity}(\mathcal{H}(z)) \longrightarrow 0$$

in probability (and typically exponentially fast in $|\mathcal{S}_i|$). This completes the proof. $\qquad\square$

## E    Variance reduction and error bounds for the ensemble anomaly score $\mathcal{I}$

To demonstrate variance reduction (proof for Eq. 14) for the ensemble isolation score, we proceed in the following steps:

**Lemma E.1** (Variance Bound for Individual Isolation Scores). *For a single subset $\mathcal{S}_i$, the variance of the base anomaly score $\mathcal{F}(x, \mathcal{S}_i)$ is bounded by:*

$$\text{Var}(\mathcal{F}(x, \mathcal{S}_i)) \leq \frac{1}{4}.$$

*Proof.* The base anomaly score $\mathcal{F}(x, \mathcal{S}_i)$ is a random variable bounded in the interval $[0, 1]$. For any random variable $X$ with values in $[a, b]$, the variance is bounded by:

$$\text{Var}(X) \leq \frac{(b-a)^2}{4}.$$

Here, $a = 0$ and $b = 1$, so:

$$\text{Var}(\mathcal{F}(x, \mathcal{S}_i)) \leq \frac{(1-0)^2}{4} = \frac{1}{4}.$$

Thus, the variance of the base anomaly score is bounded as claimed. $\qquad\square$

**Lemma E.2** (Variance Reduction for Ensemble Isolation Score). *Let the ensemble isolation score $I(x)$ be the average of $n$ independent base anomaly scores:*

$$I(x) = \frac{1}{n} \sum_{i=1}^{n} \mathcal{F}(x, \mathcal{S}_i).$$

*Then, the variance of $I(x)$ satisfies:*

$$\text{Var}(I(x)) \leq \frac{1}{4n}.$$

*Proof.* Let $\mathcal{F}_i = \mathcal{F}(x, \mathcal{S}_i)$ for simplicity. The variance of the average of $n$ independent random variables is given by:

$$\text{Var}\left(\frac{1}{n} \sum_{i=1}^{n} \mathcal{F}_i\right) = \frac{1}{n^2} \sum_{i=1}^{n} \text{Var}(\mathcal{F}_i).$$

From Lemma E.1, we know that $\text{Var}(\mathcal{F}_i) \leq \frac{1}{4}$ for all $i$. Substituting this bound:

$$\text{Var}(I(x)) = \frac{1}{n^2} \sum_{i=1}^{n} \text{Var}(\mathcal{F}_i) \leq \frac{1}{n^2} \sum_{i=1}^{n} \frac{1}{4}.$$

Simplify the summation:

$$\text{Var}(I(x)) \leq \frac{1}{n^2} \cdot n \cdot \frac{1}{4} = \frac{1}{4n}.$$

Thus, the variance of the ensemble isolation score is bounded by $\frac{1}{4n}$ as stated in Eq. 14 in the main paper. $\qquad\square$

### Error Bound via Bernstein's Inequality Boucheron et al. [2013]

**Lemma E.3** (Error Bound). *Define the average variance $\nu^2 = \frac{1}{n}\sum_{i=1}^n \mathrm{Var}(\mathcal{F}_i)$, which satisfies $\nu^2 \leq \frac{1}{4}$. Then, for any $\epsilon > 0$, the probability that $I(x)$ deviates from its expected value $\mathbb{E}[I(x)]$ by at least $\epsilon$ is bounded by:*

$$P\left(|I(x) - \mathbb{E}[I(x)]| \geq \epsilon\right) \leq 2\exp\left(-\frac{n\epsilon^2}{\frac{1}{2} + \frac{2}{3}\epsilon}\right).$$

*Proof.* We apply Bernstein's inequality for independent random variables. Let $\mathcal{F}_i$ be independent with mean $\mu_i = \mathbb{E}[\mathcal{F}_i]$, variance $\nu_i^2 = \mathrm{Var}(\mathcal{F}_i)$, and bounded range $[0, 1]$. The inequality states:

$$P\left(\left|\sum_{i=1}^n (\mathcal{F}_i - \mu_i)\right| \geq n\epsilon\right) \leq 2\exp\left(-\frac{n^2\epsilon^2}{2\sum_{i=1}^n \nu_i^2 + \frac{2}{3}n\epsilon}\right).$$

Substituting $I(x)$ and scaling by $\frac{1}{n}$:

$$P\left(|I(x) - \mathbb{E}[I(x)]| \geq \epsilon\right) \leq 2\exp\left(-\frac{n\epsilon^2}{2\nu^2 + \frac{2}{3}\epsilon}\right),$$

where $\nu^2 = \frac{1}{n}\sum_{i=1}^n \nu_i^2$ is the average variance. Using Lemma E.1, we know $\nu_i^2 \leq \frac{1}{4}$, so $\nu^2 \leq \frac{1}{4}$. Substituting this bound:

$$P\left(|I(x) - \mathbb{E}[I(x)]| \geq \epsilon\right) \leq 2\exp\left(-\frac{n\epsilon^2}{\frac{1}{2} + \frac{2}{3}\epsilon}\right).$$

Therefore, the probability of large deviations from the expected isolation score decreases exponentially with the number of hyperspheres $n$ as stated in Eq. 15 in the main paper. $\square$

## F   Proof of Theorem 3.15

We prove that under the $\delta$-separation assumption, a normal point $x$ obtains a small isolation score $I(x)$, whereas a "typical" anomaly $z$ obtains a large isolation score.

*Proof.* We break the proof into two parts: we first prove Lemma 3.13, then analyze the **ensemble** average over $n$ subsets.

**Part A: Proof for Lemma 3.13**

By the $\delta$-separation assumption (Definition 3.1), there exist radii $\sigma, \delta > 0$ and small $\varepsilon, \varepsilon' \in (0, 1)$ such that:

1. **Normal-Point Proximity**: With probability $\geq 1 - \varepsilon$ over the draw of a normal point $x \sim P_\mathcal{N}$, there exists at least one center $\mu_j$ satisfying $\|x - \mu_j\| \leq \sigma$.

2. **Anomaly Exclusion**: With probability $\geq 1 - \varepsilon'$ over the draw of an anomaly $z \sim P_\mathcal{A}$, we have $\|z - \mu_j\| \geq \sigma + \delta$ for *all* $j$.

Let $\omega$ be the size of each sampled subset, and let $\{S_i\}_{i=1}^n$ be the i.i.d. subsets. Consider one such subset $S$. Denote by

$$\mathcal{E} = \text{"event that } S \text{ includes at least one normal point from each cluster,"}$$

whereby "each cluster" we informally refer to each center $\mu_j$ with nontrivial normal mass. More precisely, define $C_j := \{x : \|x - \mu_j\| \leq \sigma\}$, and let

$$\mathcal{E} = \{S : S \cap C_j \neq \varnothing \text{ for all relevant } j\}.$$

We can bound $\Pr(\mathcal{E})$ away from zero if $\omega$ is large enough relative to the number of clusters (and using the fact that normal points constitute an $\alpha$-fraction of the data). For instance, by the binomial bound, one gets

$$\Pr(\mathcal{E}) \geq 1 - \delta_0,$$

for some small $\delta_0$. Below, we condition on $\mathcal{E}$ and also condition on the event that the chosen point $x \sim P_{\mathcal{N}}$ satisfies the normal-proximity property (probability $\geq 1 - \varepsilon$) or that $z \sim P_{\mathcal{A}}$ satisfies the anomaly-exclusion property (probability $\geq 1 - \varepsilon'$).

### A1. Normal points obtain small $\mathcal{F}$

Fix a normal point $x$. Suppose (i) $x$ lies within $\sigma$ of some center $\mu_j$, and (ii) the event $\mathcal{E}$ holds for $\mathcal{S}$. Because $\mathcal{S} \cap C_j \neq \varnothing$, there is at least one point $y \in \mathcal{S}$ also within that same cluster. In fact:

- If $y$ is paired with another point $y'$ in the **same** cluster, the radius $\frac{1}{2}\|y - y'\| \leq \sigma$. This hypersphere is dense and encloses $x$ near its center, so $\mathrm{Pitch}(x) < 1$ and $\mathrm{NDensity} \approx 1$. Hence, the weighted distance $\mathrm{WPitch}(x) \approx 0$.

- If $y$ is instead paired with a point outside the cluster, the resulting hypersphere might be larger. However, $x$ still typically sits closer to the center than does any far-away anomaly, implying $\mathrm{WPitch}(x)$ remains relatively small compared to that of an anomaly.

Among **all** hyperspheres covering $x$, we select the "smallest cover," i.e., the one that minimizes $\mathrm{WPitch}(x)$. Thus

$$\mathcal{F}(x, \mathcal{S}) \;=\; \min_{\mathcal{H} \ni x} \mathrm{WPitch}(x, \mathcal{H}) \;<\; 1.$$

Formally, one can show that, conditioned on $\mathcal{E}$ and on the event $\|x - \mu_j\| \leq \sigma$, we have $\mathcal{F}(x, \mathcal{S}) \leq \rho$ for some small $\rho < 1$. Overall, the probability that $\mathcal{F}(x, \mathcal{S}) \leq \rho$ is at least

$$(1 - \varepsilon) \times (1 - \delta_0) \;\geq\; 1 - \big(\varepsilon + \delta_0\big).$$

### A2. Anomalies obtain large $\mathcal{F}$

Now fix an anomaly $z$. Suppose $\|z - \mu_j\| \geq \sigma + \delta$ for all $j$ (probability $\geq 1 - \varepsilon'$). Two cases arise:

1. $z$ **not in** $\mathcal{S}$.
   If a normal-centered hypersphere covers $z$, that hypersphere has radius $\leq \sigma$. Since $z$ stands at distance $\geq \delta$ from the center, $\mathrm{Pitch}(z) \approx 1$. Otherwise, if $z$ is not covered, $\mathcal{F}(z, \mathcal{S}) = 1$ by definition.

2. $z$ **in** $\mathcal{S}$.
   A hypersphere centered on $z$ is sparse (few neighbors), thus $\mathrm{NDensity} \approx 0$. Hence $\mathrm{WPitch}(z) \approx \mathrm{Pitch}(z) \approx 1$.

In both scenarios, no hypersphere yields $\mathrm{WPitch} < 1$. Consequently,

$$\mathcal{F}(z, \mathcal{S}) \;\geq\; \gamma$$

for some $\gamma$ close to 1. The probability of this event is at least $1 - \varepsilon'$. Thus, for a typical anomaly $z$, $\mathcal{F}(z, \mathcal{S}) \approx 1$ occurs with probability $\geq 1 - \varepsilon'$.

### Part B: Ensemble averaging

Since the subsets $\mathcal{S}_1, \ldots, \mathcal{S}_n$ are i.i.d. uniform samples, the probability that each $\mathcal{E}$ is high. Even if *some* subsets fail, the *majority* will cleanly separate normal points from anomalies. Formally:

- **Expected base score.** Define

$$\kappa_N \;=\; \mathbb{E}[\mathcal{F}(x, \mathcal{S}) \mid x \sim P_{\mathcal{N}}], \quad \kappa_A \;=\; \mathbb{E}[\mathcal{F}(z, \mathcal{S}) \mid z \sim P_{\mathcal{A}}],$$

  By the above arguments, $\kappa_N < \kappa_A \leq 1$. Typically $\kappa_N \approx 0$ and $\kappa_A \approx 1$.

- **Final isolation score.** For each point $x$, define

$$I(x) \;=\; \frac{1}{n}\sum_{i=1}^{n}\mathcal{F}(x,\mathcal{S}_i).$$

By linearity of expectation, $\mathbb{E}[I(x) \mid x \in P_\mathcal{N}] = \kappa_N$ and $\mathbb{E}[I(x) \mid x \in P_\mathcal{A}] = \kappa_A$. Hence $\kappa_N < \kappa_A$ implies that normal points will, on average, have lower isolation scores than anomalies.

- **Variance and concentration.** Each $\mathcal{F}(\cdot,\mathcal{S}_i)$ takes values in $[0,1]$, so $\mathrm{Var}(\mathcal{F}) \leq \frac{1}{4}$. By a standard variance-addition or concentration bound (e.g., Bernstein's inequality), the averaging over $n$ subsets yields $\mathrm{Var}[I(x)] \leq \frac{1}{4n}$. Thus, with high probability over the sampling of $\mathcal{S}_i$, the isolation scores for normal points cluster around $\kappa_N$ and for anomalies around $\kappa_A$, creating a clear separation as $n$ grows.

---

Conclusion: $I(\text{normal}) \approx \kappa_N < \kappa_A \approx I(\text{anomaly})$     with high probability.

$\square$

## G   Algorithmic details

In general, ADERH separates anomalies from regular samples in a two-step process (see Algo. 1).

**Step I**. In the first step, ADERH generates a set of n subsets by performing uniform random sampling with replacement, denoted as $\mathrm{SUBSETS}(\mathcal{D}, \mathrm{n}, \omega) = \{\mathcal{S}_1, \ldots, \mathcal{S}_n\}$. For each $\mathcal{S}_i \in \mathrm{SUBSETS}$ and all elements $x \in \mathcal{S}_i$, a random partner with $y \in \mathcal{S}_i$ is determined. Subsequently, two hyperspheres with the centers $x$ and $y$ are formed. We consider an ensemble of hyperspheres with varying radii, where ADERH estimates key properties such as the density of each hypersphere under the empirical data distribution.

**Step II**. ADERH utilizes an ensemble of hyperspheres across all $S_i \in \mathrm{SUBSETS}$ to minimize variance and reduce deviations in the computed anomaly scores, enhancing robustness. For each data point and each subset $\mathcal{S}_i \in \mathrm{SUBSETS}$, the algorithm determines the smallest cover $\mathrm{SC}(x, \mathcal{S}_i)$. By considering the positions of the data points and the densities of the ensemble of hyperspheres, ADERH calculates the anomaly score based on the smallest covers across the different subsets. If a data point consistently receives high anomaly scores from the ensemble of hyperspheres, it can be confidently identified as an anomaly.

## H   Parameter setting

In ADERH, we fix two principal parameters: (i) $n$, the number of random subsets (the ensemble size), and (ii) $\omega$, the size of each random subset. These choices mirror the logic behind Isolation Forest (IForest) Liu et al. [2008], which typically employs about 100–300 estimators (trees) and often uses up to 256 samples per tree. We adopt $n = 256$ in a similar spirit: increasing the ensemble size reduces variance (see Appendix E), but we observe diminishing returns beyond 200–300 subsets in practice (Fig. 4b).

Where IForest allocates 256 data points to each tree, ADERH requires far fewer points per subset, and we set $\omega = 18$. The rationale is that ADERH does not rely on hierarchical splits but rather forms hyperspheres from pairs of points in these small subsets. Smaller subsets risk underrepresenting normal structure, while larger subsets incur higher contamination probability (since more anomalies might appear among the centers) and yield limited gains in AUC-ROC or AUC-PR (Fig. 4a). Consequently, $\omega = 18$ balances computational efficiency with coverage of the underlying data patterns, allowing ADERH's hyperspheres to remain compact and centered around typical (normal) samples.

**Competitor's parameter**   For competitors, we used the default parameter settings as specified in the respective papers (Table 7).

---

**Algorithm 1:** ADERH

---

**input** : dataset: $\mathcal{D}$,
        # of subsets: n,
        subset size: $\omega$

**output** : vector containing the anomaly score for all samples $I$

```
// At the beginning, all anomaly scores are initialized with 0.
```
1   $\mathcal{I} := \vec{0}$
```
// First, n subsets are generated via random sampling with replacement, where each subset contains ω
   samples (Definition 3.2).
```
2   initialize $\mathrm{SUBSETS}(\mathcal{D}, \mathrm{n}, \omega)$
3   $\mathcal{E}_{\mathrm{all}} := \emptyset$
4   **for** $\mathcal{S}_i \in \mathrm{SUBSETS}(\mathcal{D}, \mathrm{n}, \omega)$ **do**
5      $\mathcal{E}(\mathcal{S}_i) := \emptyset$
6      **for** $x \in \mathcal{S}_i$ **do**
```
         // Generate the hyperspheres of the data point x and its random partner according to
            Definition 3.4
```
7          $\mathcal{H} := \mathcal{H}(x, \mathcal{S}_i)$ // Hypersphere with center $x$
8          $\mathcal{H}' := \mathcal{H}(P(x, \mathcal{S}_i), \mathcal{S}_i)$ // Hypersphere with center $P(x, \mathcal{S}_i)$
```
         // Determine the density of the created hyperspheres (Definition 3.7)
```
9          $\mathrm{Density}(\mathcal{H}) := \frac{|X_{\mathcal{H}} \cap \mathcal{D}|}{\mathrm{R}(\mathcal{H})}$
10         $\mathrm{Density}(\mathcal{H}') := \frac{|X_{\mathcal{H}'} \cap \mathcal{D}|}{\mathrm{R}(\mathcal{H}')}$
```
         // Add the two hyperspheres to the ensemble (Definition 3.5)
```
11         $\mathcal{E}(\mathcal{S}_i) := \mathcal{E}(\mathcal{S}_i) \cup \{\mathcal{H}, \mathcal{H}'\}$
12      **end**
```
      // Normalize the densities of the hyperspheres according to Definition 3.8
```
13      **for** $\mathcal{H} \in \mathcal{E}(\mathcal{S}_i)$ **do**
14         $\mathrm{NDensity}(\mathcal{H}, \mathcal{S}_i) := \frac{\mathrm{Density}(\mathcal{H})}{max_{\mathcal{H}_j \in \mathcal{E}(\mathcal{S}_i)} \mathrm{Density}(\mathcal{H}_j)}$
15      **end**
16      $\mathcal{E}_{\mathrm{all}} := \mathcal{E}_{\mathrm{all}} \cup \mathcal{E}(\mathcal{S}_i)$
17   **end**
18   **for** $\mathcal{E}(\mathcal{S}_i) \in \mathcal{E}_{all}$ **do**
19      **for** $y \in \mathcal{D}$ **do**
```
         // If the data point is covered by at least one hypersphere H ∈ E(Sᵢ), the isolation value is
            evaluated according to Definition 3.14. If the data point y is not covered by any
            hypersphere, then it receives the maximum value of 1.
```
20         $I(y) := I(y) + \frac{1}{n} \mathcal{F}(y, \mathcal{S}_i)$
21      **end**
22   **end**

---

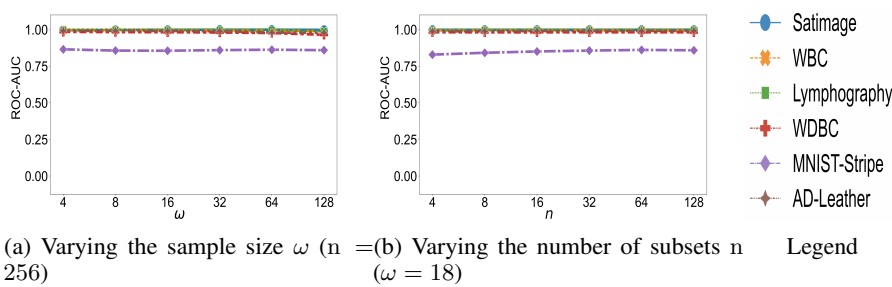

(a) Varying the sample size $\omega$ (n = 256)

(b) Varying the number of subsets n ($\omega = 18$)

Legend

Figure 4: Stability comparison of ADERH by increasing the sample size $\omega$ and number of subsets n.

**Grid search experiment**    In addition to the default settings, we conducted a grid search experiment to further investigate performance by exploring various parameter configurations for both ADERH and the competitors (Appendix Q).

## I   Robustness

In the following, we investigate the behavior of ADERH with regard to parameter stability. In the first experiment (Fig. 4), we increase the sample size $\omega$ of a random subset $\mathcal{S}_i$ ($|\mathcal{S}_i| = \omega$). As shown in Fig. 4a, ADERH generally attains high AUC-ROC values across varying cardinalities $\omega$ of the random subset $\mathcal{S}_i$. In the second experiment (Fig. 4b), we increase the number of subsets n. As

expected, slightly better values are achieved when the number of subsets increases. As n increases, more estimates can be made, improving anomaly detection accuracy. Moreover, these observations align with our findings on variance reduction. By increasing n, we effectively reduce the variance over all base anomaly scores and increase robustness and, according to the Bernstein inequality, lead to tighter bounds and greater reliability in distinguishing anomalies.

## J  Datasets

Details regarding the used datasets are given in Table 6.

Table 6: Statistics of the used datasets

| Dataset | # Instances | # Dimensions | # Anomalies (%) |
|---|---|---|---|
| Optdigits | 5216 | 64 | 150(0.0288) |
| Wbc | 223 | 9 | 10(0.0448) |
| Lymphography | 148 | 18 | 6(0.0405) |
| Celeba | 202599 | 39 | 4547(0.0224) |
| Skin | 245057 | 3 | 50859(0.2075) |
| Pendigits | 6870 | 16 | 156(0.0227) |
| Wdbc | 367 | 30 | 10(0.0272) |
| AD-Toothbrush | 10000 | 512 | 500(0.0500) |
| Wpbc | 198 | 33 | 47(0.2374) |
| AD-Leather | 10000 | 512 | 500(0.0500) |
| Satimage-2 | 5803 | 36 | 71(0.0122) |
| Backdoor | 95329 | 196 | 2329(0.0244) |
| MNIST-C-Stripe | 10000 | 512 | 500(0.0500) |
| Waveform | 3443 | 21 | 100(0.0290) |
| Cardio | 1831 | 21 | 176(0.0961) |
| AD-Bottle | 10000 | 512 | 500(0.0500) |
| Census | 299285 | 500 | 18568(0.0620) |
| Wine | 129 | 13 | 10(0.0775) |
| Musk | 3062 | 166 | 97(0.0317) |

## K  Experimental details

Experiments were conducted on an Intel Core i7-10700K, 3.8 GHz, 32 GB RAM, with runtime averaged over ten consecutive runs. Real-world datasets were sourced from the AdBenchmark repository Han et al. [2022], with MNIST-Variation and AD-Variation datasets derived using ResNet18 features pre-trained on ImageNet Han et al. [2022]. Table 6 summarizes dataset statistics. Implementations were obtained from Zhao et al. [2019b], Xu et al. [2023a].

We now detail the common experimental procedures used across *all* experiments in this paper:

1. **Datasets and Preprocessing.** We consider $\mathcal{D}$ real-world datasets, each containing a mixture of normal and anomalous points. To maintain consistency, we normalize all datasets to the range $[0, 1]$ using the MinMaxScaler Pedregosa et al. [2011]. A summary of each dataset's statistics is provided in Table 6.

2. **Train/Test Splitting.** We employ a stratified split with 70% of the data for training and 30% for testing, ensuring that both splits maintain the same proportion of anomalies. This split is repeated three times with different random seeds; we report the *average* performance metrics (AUC-ROC and AUC-PR) across these three runs.

3. **Evaluation Metrics.** We adopt the AUC-ROC and AUC-PR Davis and Goodrich [2006] as our primary metrics, as they are widely used and provide stable comparisons for imbalanced datasets. We also conduct a paired Wilcoxon signed-rank test with Holm–Bonferroni correction McDonald [2014] to determine statistical significance.

4. **Implementation Details.** All methods (including ours) are implemented in Python, and we use the public repositories Zhao et al. [2019b], Xu et al. [2023a] for baseline implementations

Table 7: Default parameter setting.

| Algorithm | Description | Set |
|---|---|---|
| ADERH | n | {256} |
| | $\omega$ | {18} |
| INNE | $\#estimators$ | {200} |
| | $\#maxsamples$ | {8} |
| Isolation Forest | $\#estimators$ | {100} |
| | $\#maxsamples$ | {256} |
| | $maxfeatures$ | {1.0} |
| EIF | $extensionlevel$ | {1} |
| | $\#maxsamples$ | {256} |
| | $\#estimators$ | {100} |
| DIF | $\#ensemble$ | {6} |
| | $\#estimators$ | {100} |
| | $\#maxsamples$ | {256} |
| PIDForest | $maxdepth$ | {10} |
| | $\#trees$ | {20} |
| | $\#samples$ | {256} |
| LOF | $MinPts$ | {5} |
| DeepSVDD | $epochs$ | {100} |
| | $batchsize$ | {32} |
| | $dropout$ | {0.2} |
| OCSVM | $kernel$ | {$RBF$} |
| | $degree$ | {3} |
| | $tol$ | {$1e^{-3}$} |
| | $nu$ | {0.5} |
| RCA | $epochs$ | {100} |
| | $batchsize$ | {64} |
| | $lr$ | {$1e^{-3}$} |
| | $repDim$ | {128} |
| RDP | $epochs$ | {100} |
| | $batchsize$ | {64} |
| | $lr$ | {$1e^{-3}$} |
| | $prt\_steps$ | {10} |
| LODA | $\#bins$ | {10} |
| | $\#randomcuts$ | {100} |
| SLAD | $epochs$ | {100} |
| | $batchsize$ | {128} |
| | $lr$ | {$1e^{-3}$} |
| | $n\_slad\_ensemble$ | {20} |
| | $subspace\_pool\_size$ | {50} |

Table 8: This table presents the results of all algorithms using the default parameters outlined in the original paper. Hereby, the best values are shown in bold, and the runner-up is underlined. The 'AVG Rank' row of the table lists the average rank achieved by all algorithms in the metric AUC-PR

| Dataset | ADERH | INNE | IForest | EIF | DIF | PIDForest | LOF | DeepSVDD | RCA | RDP | OCSVM | LODA | SLAD | DTE | UniCAD |
|---|---|---|---|---|---|---|---|---|---|---|---|---|---|---|---|
| Optdigits | 0.061 (4) | 0.064 (3) | 0.049 (6) | 0.050 (5) | 0.037 (8) | 0.029 (11) | 0.065 (2) | 0.028 (13) | **0.069 (1)** | 0.030 (10) | 0.029 (11) | 0.027 (14) | 0.034 (9) | 0.045 (7) | 0.027 (14) |
| Wbc | **1.000 (1)** | 0.342 (11) | 0.994 (4) | **1.000 (1)** | 0.120 (14) | 0.759 (8) | 0.237 (12) | 0.359 (10) | 0.935 (6) | 0.519 (9) | **1.000 (1)** | 0.972 (5) | 0.117 (13) | 0.197 (13) | 0.914 (7) |
| Lymphography | **1.000 (1)** | 0.811 (10) | 0.978 (6) | **1.000 (1)** | 0.399 (13) | 0.841 (9) | **1.000 (1)** | 0.543 (12) | **1.000 (1)** | 0.844 (7) | **1.000 (1)** | 0.242 (15) | 0.617 (11) | 0.266 (14) | 0.843 (8) |
| Celeba | 0.060 (5) | 0.044 (9) | 0.060 (5) | 0.065 (4) | 0.053 (8) | 0.055 (7) | 0.018 (14) | 0.037 (12) | 0.028 (13) | 0.028 (13) | 0.076 (2) | 0.040 (11) | 0.068 (3) | 0.000 | **0.109 (1)** |
| Skin | 0.345 (2) | 0.286 (7) | 0.256 (10) | 0.273 (8) | 0.258 (9) | 0.289 (6) | 0.238 (11) | 0.196 (12) | 0.291 (5) | **0.372 (1)** | 0.187 (13) | 0.185 (14) | 0.326 (3) | 0.000 | 0.306 (4) |
| Pendigits | **0.309 (1)** | 0.179 (9) | 0.305 (2) | 0.267 (5) | 0.282 (4) | 0.210 (7) | 0.038 (13) | 0.018 (15) | 0.105 (12) | 0.121 (11) | 0.226 (6) | 0.289 (3) | 0.198 (8) | 0.035 (14) | 0.174 (10) |
| Wdbc | 0.614 (4) | 0.315 (10) | 0.613 (5) | 0.692 (2) | 0.116 (14) | 0.446 (7) | 0.484 (6) | 0.169 (12) | 0.354 (9) | 0.230 (11) | **0.714 (1)** | 0.636 (3) | 0.130 (13) | 0.095 (15) | 0.396 (8) |
| AD-Toothbrush | 0.840 (2) | 0.828 (4) | 0.809 (5) | 0.793 (6) | 0.836 (3) | 0.290 (15) | 0.630 (11) | 0.789 (7) | 0.587 (13) | 0.768 (8) | 0.676 (10) | 0.597 (12) | **0.898 (1)** | 0.460 (14) | 0.767 (9) |
| Wpbc | 0.261 (3) | 0.253 (8) | 0.239 (13) | 0.247 (11) | 0.228 (15) | 0.248 (10) | 0.258 (5) | 0.249 (9) | 0.259 (4) | 0.246 (12) | 0.235 (14) | **0.271 (1)** | 0.265 (2) | 0.254 (6) | 0.254 (6) |
| AD-Leather | **0.975 (1)** | 0.688 (12) | 0.953 (3) | 0.953 (3) | 0.919 (6) | 0.252 (15) | 0.552 (13) | 0.903 (7) | 0.846 (10) | 0.934 (5) | 0.863 (9) | 0.726 (11) | 0.957 (2) | 0.551 (14) | 0.898 (8) |
| Satimage-2 | 0.957 (2) | 0.854 (8) | 0.921 (5) | 0.933 (4) | 0.761 (9) | 0.699 (10) | 0.030 (15) | 0.042 (14) | 0.938 (3) | 0.394 (11) | 0.862 (7) | 0.913 (6) | 0.134 (12) | 0.076 (13) | **0.960 (1)** |
| MNIST-C-Stripe | 0.699 (2) | 0.429 (9) | 0.542 (6) | 0.604 (5) | 0.614 (4) | 0.050 (14) | 0.046 (15) | 0.117 (12) | **0.752 (1)** | 0.339 (11) | 0.528 (7) | 0.676 (3) | 0.522 (8) | 0.070 (13) | 0.356 (10) |
| Shuttle | 0.918 (5) | 0.728 (8) | **0.977 (1)** | 0.965 (2) | 0.573 (11) | 0.652 (10) | 0.095 (14) | 0.106 (13) | 0.955 (4) | 0.692 (9) | 0.959 (3) | 0.211 (12) | 0.864 (7) | 0.000 | 0.904 (6) |
| Waveform | 0.144 (2) | 0.134 (3) | 0.057 (8) | 0.060 (7) | 0.073 (5) | 0.038 (12) | 0.111 (4) | 0.038 (12) | 0.062 (6) | 0.046 (10) | 0.034 (14) | 0.043 (11) | **0.342 (1)** | 0.033 (15) | 0.055 (9) |
| Cardio | **0.588 (1)** | 0.449 (10) | 0.539 (5) | 0.560 (3) | 0.542 (4) | 0.387 (12) | 0.192 (13) | 0.184 (14) | 0.460 (9) | 0.499 (6) | 0.567 (2) | 0.437 (11) | 0.496 (8) | 0.155 (15) | 0.499 (6) |
| AD-Bottle | **0.940 (1)** | 0.835 (10) | 0.915 (5) | 0.913 (6) | 0.933 (4) | 0.216 (15) | 0.853 (9) | 0.829 (12) | 0.803 (13) | 0.938 (2) | 0.834 (11) | 0.912 (7) | 0.936 (3) | 0.448 (14) | 0.911 (8) |
| Census | 0.075 (2) | 0.056 (13) | 0.071 (5) | **0.076 (1)** | 0.067 (8) | 0.063 (9) | 0.063 (9) | 0.071 (5) | 0.072 (4) | 0.075 (2) | 0.062 (11) | 0.058 (12) | 0.069 (7) | 0.000 | 0.000 |
| Wine | 0.262 (3) | 0.211 (9) | 0.216 (7) | 0.237 (5) | 0.095 (13) | 0.000 (15) | 0.360 (2) | 0.257 (4) | 0.075 (14) | 0.166 (12) | 0.126 (10) | 0.217 (6) | 0.215 (8) | 0.106 (11) | **0.448 (1)** |
| Musk | **1.000 (1)** | **1.000 (1)** | 0.964 (6) | 0.945 (7) | 0.691 (9) | 0.991 (4) | 0.037 (15) | 0.207 (11) | 0.750 (8) | 0.166 (12) | 0.085 (13) | 0.224 (10) | 0.981 (5) | 0.041 (14) | **1.000 (1)** |
| AVG Rank | 2.26 | 8.11 | 5.63 | 4.53 | 8.47 | 10.32 | 9.68 | 11.26 | 6.47 | 8.63 | 7.68 | 8.79 | 6.63 | 13.21 | 6.89 |
| p-value | NA | 0.00228567 (+) | 0.0171525 (+) | 0.02369850 (+) | 0.00005341 (+) | 0.00005341 (+) | 0.00063475 (+) | 0.00005341 (+) | 0.02024470 (+) | 0.00284356 (+) | 0.02089202 (+) | 0.00035406 (+) | 0.02368190 (+) | 0.00005341 (+) | 0.02368190 (+) |

The values marked with † indicate that an error occurred during execution.

where available. Where randomness is involved, we run each method using five different random seeds $[0, 1, 2, 100, 1000]$ and average the metrics to ensure robustness.

**Performance tables** Tables 1, 8, 9, and 11 reporting our results, the best-performing method on each dataset is shown in bold, and the second-best is underlined. The "AVG Rank" row presents the mean rank of every algorithm, where a lower rank denotes stronger performance overall. The last row shows p-values from the Wilcoxon signed-rank test (at $\alpha = 0.05$) comparing our method (ADERH) to each reference approach. Here, a plus sign "(+)" denotes that ADERH achieves a statistically significant improvement.

## L   Additional results AUC-PR

Below, we present the additional AUC-PR results for all evaluated methods under their default hyperparameter settings (Table 8). As in the main paper's AUC-ROC comparison, ADERH consistently achieves top rankings across most datasets in AUC-PR. Notably, ADERH attains the best AUC-PR scores on 7 datasets and second-best on 6, yielding the lowest average rank of 2.26 among all competing methods. These results underscore ADERH's robust performance, particularly in unbalanced scenarios where the AUC-PR metric is more sensitive to class imbalance and rare anomalies. Similar to our AUC-ROC findings, the paired Wilcoxon signed-rank tests indicate that ADERH's improvements over baseline methods are statistically significant. By forming pairs of hyperspheres with diverse radii and integrating Pitch-based boundary detection alongside NDensity-driven hypersphere weighting, ADERH consistently achieves superior precision-recall performance relative to both traditional isolation-based and deep-learning-based anomaly detectors across a broad range of real-world datasets.

## M   Runtime complexity

As described in Section G, the operation of ADERH consists of two distinct steps. The computational complexity associated with each step of ADERH is explained before these findings are summarized to determine the total runtime complexity of ADERH. Initially, the algorithm generates a set of subsamples $\text{SUBSETS}(\mathcal{D}, n, \omega)$ from a dataset $\mathcal{D}$, where the number of subsamples equals n. For each individual set $\mathcal{S}_i \in \text{SUBSETS}(\mathcal{D}, n, \omega)$, the cardinality is equal to $\omega$. A pair of hyperspheres is constructed for each element $x \in \mathcal{S}_i$. Thus, the generation of the hyperspheres has a complexity of $O(2\,n\,\omega)$. In addition, the density must be determined for each of the hyperspheres. This operation has a $O(m)$ complexity regarding each hypersphere, where $m = |\mathcal{D}|$. This results in a total complexity of $O(2\,n\,\omega\,m)$ for the first step. For every data point within $\mathcal{D}$ and each element in the collection $\mathcal{E}_{\text{all}} = \{\mathcal{E}(\mathcal{S}_1), \ldots, \mathcal{E}(\mathcal{S}_n)\}$, the algorithm identifies the smallest covering hypersphere (SC). Given that the size of $\mathcal{E}_{\text{all}}$ equals n and each $\mathcal{E}(\mathcal{S}_i) \in \mathcal{E}_{\text{all}}$ contains $2\omega$ hyperspheres, finding SC for all data points across all hyperspheres has a computational complexity of $O(2\,n\,\omega\,m)$. By neglecting the constant factors, we obtain a combined runtime complexity of $O(n\,\omega\,m)$. This means that ADERH

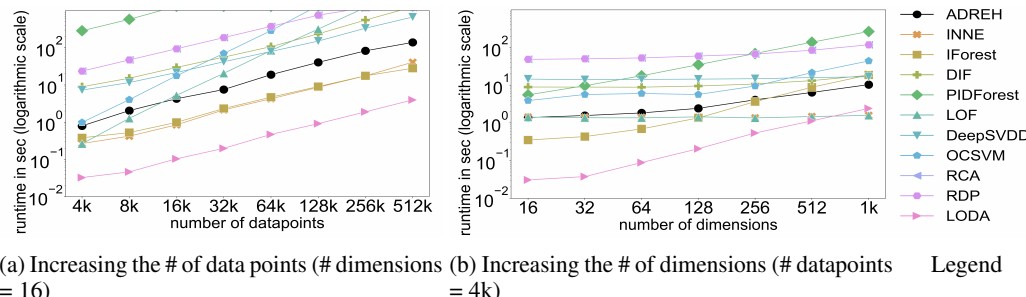

(a) Increasing the # of data points (# dimensions = 16)

(b) Increasing the # of dimensions (# datapoints = 4k)

Legend

Figure 5: Runtime experiment using a dataset consisting of Gaussian distributed regions with uniformly distributed anomalies around the Gaussian regions.

has the same asymptotic behavior as, for example, INNE. As the runtime of ADERH scales linearly with m, $\omega$, and n, the method is efficient and well-suited for large datasets.

## N    Runtime evaluation

In this experiment, we analyze the scalability of ADERH and its competitors, particularly other isolation-based approaches. For this purpose, we created Gaussian-distributed regions with uniformly distributed anomalies around them. The default parameter settings, as described in the respective papers, were used for all algorithms (Table 7). In the first experiment, the number of data points, denoted as m, is progressively increased while keeping the number of dimensions fixed at $d = 16$ (Fig. 5a). The results demonstrate that the runtime of isolation-based methods scales linearly as m increases. ADERH exhibits a similar asymptotic runtime behavior as INNE or IForest, but with a consistently higher runtime by a constant factor (as discussed in Section M). LOF and deep learning-based anomaly detection methods show poor scalability and are, therefore, only partially suitable for large amounts of data. In the second experiment, the dimensions $d$ of the data points are increased with the number of samples fixed at m = $4k$ (Fig. 5b). In this case as well, ADERH demonstrates high scalability, performing comparably to other state-of-the-art approaches.

## O    Ablation study: different settings of ADERH

In this ablation study, we systematically compare the proposed method ADERH under three configurations. The first configuration, called ADERH [Only Pitch], uses only the distance-based ratio $\mathrm{Pitch}$ to quantify how close a point lies to the center of a hypersphere but omits the normalized density term $\mathrm{NDensity}$. The second configuration, referred to as ADERH [Full $r$, #1 Hypersphere], generates exactly one hypersphere per pair $(x, y)$ with radius $\mathrm{dist}(x, y)$ rather than splitting it into two half-radius hyperspheres. The third configuration is the default ADERH method described in the main text, which creates two smaller hyperspheres of radius $\frac{1}{2}\mathrm{dist}(x, y)$ for each pair and incorporates both $\mathrm{Pitch}$ and $\mathrm{NDensity}$. Table 9 compares the AUC-ROC scores for all three settings. In most datasets, ADERH [Full $r$, #1 Hypersphere] yields the poorest performance because a single large-radius hypersphere often encompasses anomalies as well as normal points, obscuring their distinctions. By contrast, splitting a pairwise distance into two half-radius hyperspheres reduces the risk of covering outliers and lowers the overall variance in hypersphere sizes, thereby improving separability. The omission of $\mathrm{NDensity}$ in ADERH [Only Pitch] also impairs performance, especially when anomalies themselves become hypersphere centers. Without down-weighting sparse (anomalous) hyperspheres, anomaly scores can be inflated or misassigned. In contrast, the full ADERH method yields, on average, the highest AUC-ROC values, indicating that combining a ratio-based distance measure $\mathrm{Pitch}$ with density-aware weighting $\mathrm{NDensity}$ and forming two compact hyperspheres for each pair of data points is crucial for robust outlier isolation. Overall, these results confirm that both halving the radii by incorporating $\mathrm{NDensity}$ and $\mathrm{Pitch}$ are essential design choices in ADERH.

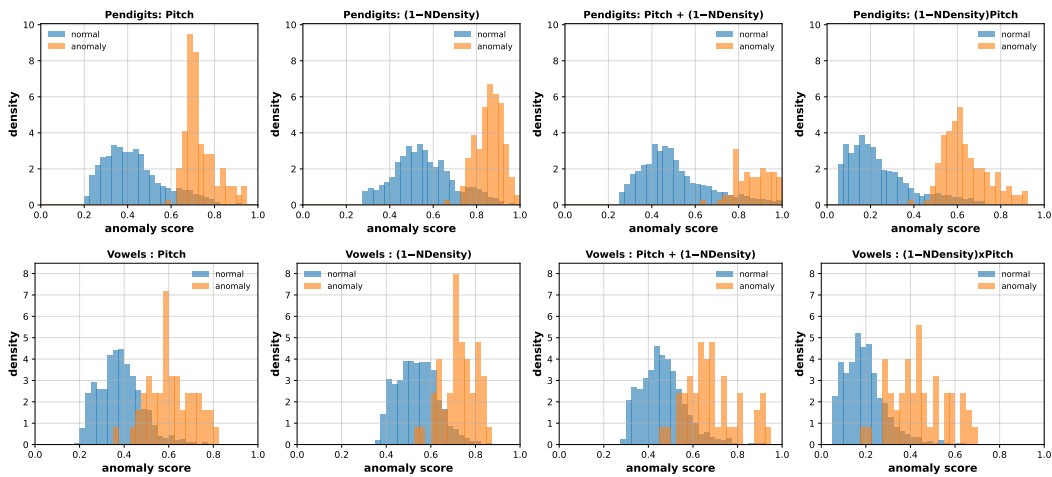

Figure 6: **Score distributions for normals vs. anomalies.** Panels show *Pitch*, $1 - \text{NDensity}$, and their *additive* vs. *multiplicative* combinations (all scores in $[0, 1]$, identical axes within each dataset). Multiplicative fusion produces the lowest normal–anomaly overlap and the most pronounced anomaly separation, aligning with our theoretical motivation and corroborating the aggregate ablation in Appendix Q.

### O.1 Visual evidence for Pitch, NDensity, and multiplicative fusion

To make the roles of $\text{Pitch}$ (boundary proximity) and $\text{NDensity}$ (local sparsity) tangible, we visualize score distributions for normals vs. anomalies on two representative datasets. Each row in Figure 6 comprises four panels: (i) Pitch, (ii) $1 - \text{NDensity}$, (iii) the additive combination $(1 - \text{NDensity}) + \text{Pitch}$, and (iv) the multiplicative combination $(1 - \text{NDensity}) \times \text{Pitch}$.

Within each dataset, scores are scaled to $[0, 1]$ and panels share identical axes to enable direct comparison. As expected, Pitch alone emphasizes boundary-adjacent instances but can elevate scores in dense regions; $1 - \text{NDensity}$ highlights sparse regions yet may pick up normal tails. The *multiplicative* fusion yields the smallest overlap between normal and anomalous distributions and the clearest right-shift of anomalies, indicating that high anomaly scores arise primarily when *both* boundary proximity and sparsity coincide.

## P  Ensembling improves anomaly detection over any single subset

Table 10 compares the proposed *ensemble* ADERH score to the *best single subset* variant across a diverse set of 19 datasets spanning tabular and visual AD benchmarks. All results follow our standard reporting protocol: we report means across repeated runs and assess across-dataset differences using a paired Wilcoxon signed-rank test with Holm correction at $\alpha=0.05$.

**Summary of results.** The ensemble outperforms the best single subset on *all* datasets (19/19 wins). Averaged over datasets, the overall advantage is statistically significant under the paired Wilcoxon test with Holm correction ($\alpha=0.05$). These results substantiate our variance-reduction motivation for ensembling: averaging scores across randomly constructed subsets mitigates idiosyncratic failure modes of any single subset and stabilizes decision boundaries across data regimes. Empirically, we observe consistent rightward shifts of anomaly-score distributions and tighter normal-score concentrations for the ensemble relative to the best single subset, aligning with our theoretical analysis on variance control.

Table 9: This table presents the AUC-ROC results of different configuration of ADERH.

| Dataset | ADERH | ADERH [Only Pitch] | ADERH [Full $r$, #1Hypersphere ] |
|---|---|---|---|
| Optdigits | **0.775 (1)** | 0.671 (3) | **0.775 (1)** |
| Wbc | **1.000 (1)** | **1.000 (1)** | 0.970 (3) |
| Lymphography | **1.000 (1)** | 0.996 (3) | **1.000 (1)** |
| Celeba | **0.732 (1)** | 0.709 (2) | 0.691 (3) |
| Skin | **0.788 (1)** | 0.781 (2) | 0.759 (3) |
| Pendigits | **0.962 (1)** | 0.960 (2) | 0.959 (3) |
| Wdbc | **0.981 (1)** | **0.981 (1)** | 0.952 (3) |
| AD-Toothbrush | **0.901 (1)** | 0.843 (3) | 0.871 (2) |
| Wpbc | **0.554 (1)** | 0.522 (3) | 0.541 (2) |
| AD-Leather | **0.991 (1)** | 0.980 (3) | 0.985 (2) |
| Satimage-2 | **0.998 (1)** | **0.998 (1)** | 0.997 (3) |
| Backdoor | 0.889 (2) | **0.898 (1)** | 0.826 (3) |
| MNIST-C-Stripe | **0.986 (1)** | 0.984 (2) | 0.976 (3) |
| Shuttle | **0.987 (1)** | **0.987 (1)** | 0.979 (3) |
| Waveform | **0.768 (1)** | 0.752 (2) | 0.701 (3) |
| Cardio | **0.938 (1)** | 0.917 (3) | 0.918 (2) |
| AD-Bottle | 0.964 (2) | 0.952 (3) | **0.969 (1)** |
| Census | 0.628 (2) | **0.637 (1)** | 0.623 (3) |
| Wine | **0.839 (1)** | 0.659 (3) | 0.822 (2) |
| Musk | **1.000 (1)** | **1.000 (1)** | **1.000 (1)** |
| AVG Rank | 1.16 | 2.11 | 2.32 |
| p-value | NA | 0.00585620 (+) | 0.00185912 (+) |

## Q    Grid search experiment for isolation and non-isolation methods

In addition to the experiments reported in Section 4—where we used default hyperparameters (see Appendix H)—we conducted a comprehensive hyperparameter *grid search* for the following isolation-based methods:

- **ADERH (proposed method)**: Varying both the ensemble size $n_{\text{esti}} \in \{100, 200, 300\}$ and the random-subset size $\omega \in \{8, 18, 24\}$.

- **INNE** Bandaragoda et al. [2014]: Varying the number of estimators $n_{\text{esti}} \in \{100, 200, 300\}$ and maximum sub-sample size $\{8, 18, 24\}$.

- **IForest** Liu et al. [2008]: Varying $n_{\text{esti}} \in \{100, 200, 300\}$ and $\{128, 256, 300\}$ max samples per tree.

- **PIDForest** Gopalan et al. [2019]: Varying $n_{\text{esti}} \in \{100, 200, 300\}$ and $\{128, 256, 300\}$ samples per tree.

- **EIF** Hariri et al. [2019]: Varying $n_{\text{esti}} \in \{100, 200, 300\}$ and $\{128, 256, 300\}$ samples per tree.

- **LOF** Breunig et al. [2000]: Varying $n_{\text{neighbor}} \in \{5, 10, 20, 30, 40, 50\}$.

- **LODA** Pevnỳ [2016]: Varying $n\_randomcuts \in \{50, 100, 200\}$ and $n\_bins \in \{5, 10, 25\}$

- **DeepSVDD** Ruff et al. [2018]: Varying $\text{batchsize} \in \{50, 100, 200\}$, l2_regularizer $\in \{5, 10, 25\}$ and dropoutrate $\in \{0.2, 0.4\}$.

- **RDP** Wang et al. [2019b]: Varying $\text{batchsize} \in \{32, 64\}$ and prt_steps $\in \{5, 10, 20, 30\}$.

- **SLAD** Xu et al. [2023b]: Varying n_ensemble $\in \{10, 20, 50\}$ and subspace_pool_size $\in \{25, 50, 100\}$.

As in Appendix K, each hyperparameter configuration uses the same data splits and evaluation protocols

Table 10: Ensemble vs. best single subset (AUC-ROC). The ensemble improves performance on every dataset.

| Dataset | Ensemble ADERH | Best Single Subset |
|---|---|---|
| Optdigits | **0.775** | 0.597 |
| Wbc | **1.000** | 0.949 |
| Lymphography | **1.000** | 0.966 |
| Celeba | **0.732** | 0.631 |
| Skin | **0.788** | 0.577 |
| Pendigits | **0.962** | 0.814 |
| Wdbc | **0.981** | 0.915 |
| AD-Toothbrush | **0.901** | 0.802 |
| Wpbc | **0.554** | 0.495 |
| AD-Leather | **0.991** | 0.900 |
| Satimage-2 | **0.998** | 0.974 |
| Backdoor | **0.889** | 0.804 |
| MNIST-C-Stripe | **0.986** | 0.917 |
| Waveform | **0.768** | 0.629 |
| Cardio | **0.938** | 0.820 |
| AD-Bottle | **0.964** | 0.881 |
| Wine | **0.839** | 0.697 |
| Musk | **1.000** | 0.855 |

Table 11: This table presents the optimal AUC-ROC results achieved under various parameter settings for ADERH, INNE, IForest, PIDForest, EIF, LOF, LODA, DeepSVDD, RDP, and SLAD.

| Dataset | ADERH | INNE | IForest | PIDForest | EIF | LOF | LODA | DeepSVDD | RDP | SLAD |
|---|---|---|---|---|---|---|---|---|---|---|
| Optdigits | 0.777 (2) | **0.849 (1)** | 0.744 (4) | 0.500 (10) | 0.737 (5) | 0.571 (8) | 0.776 (3) | 0.670 (6) | 0.502 (9) | 0.603 (7) |
| Wbc | **1.000 (1)** | 0.913 (9) | **1.000 (1)** | 0.994 (6) | **1.000 (1)** | 0.997 (5) | **1.000 (1)** | 0.931 (8) | 0.958 (7) | 0.778 (10) |
| Lymphography | **1.000 (1)** | 0.989 (5) | **1.000 (1)** | 0.984 (7) | **1.000 (1)** | **1.000 (1)** | 0.895 (10) | 0.957 (9) | 0.988 (6) | 0.959 (8) |
| Celeba | 0.747 (2) | 0.689 (5) | 0.698 (4) | 0.686 (6) | 0.721 (3) | 0.475 (10) | 0.669 (7) | 0.644 (8) | 0.586 (9) | **0.787 (1)** |
| Skin | 0.788 (2) | 0.714 (6) | 0.673 (7) | 0.727 (4) | 0.724 (5) | 0.579 (9) | 0.514 (10) | 0.642 (8) | **0.810 (1)** | 0.766 (3) |
| Pendigits | **0.963 (1)** | 0.933 (7) | 0.955 (2) | 0.940 (6) | 0.954 (3) | 0.565 (10) | 0.947 (4) | 0.599 (9) | 0.905 (8) | 0.941 (5) |
| Wdbc | 0.982 (5) | 0.948 (7) | 0.984 (3) | 0.977 (6) | 0.987 (2) | 0.984 (3) | **0.990 (1)** | 0.851 (9) | 0.869 (8) | 0.787 (10) |
| AD-Toothbrush | 0.905 (5) | 0.919 (4) | 0.877 (7) | 0.500 (10) | 0.876 (9) | 0.904 (6) | 0.926 (3) | 0.929 (2) | 0.877 (7) | **0.939 (1)** |
| Wpbc | 0.554 (2) | 0.525 (6) | 0.493 (10) | 0.520 (8) | 0.522 (7) | 0.553 (3) | 0.549 (4) | **0.620 (1)** | 0.520 (8) | 0.528 (5) |
| AD-Leather | **0.993 (1)** | 0.907 (9) | 0.986 (4) | 0.500 (10) | 0.986 (3) | 0.965 (7) | 0.961 (8) | 0.981 (5) | 0.979 (6) | 0.988 (2) |
| Satimage-2 | **0.998 (1)** | 0.997 (2) | 0.993 (4) | 0.983 (6) | 0.995 (3) | 0.828 (9) | 0.991 (5) | 0.695 (10) | 0.978 (7) | 0.953 (8) |
| Backdoor | 0.895 (2) | 0.750 (8) | 0.739 (9) | 0.500 (10) | 0.790 (6) | 0.788 (7) | 0.807 (5) | 0.894 (3) | 0.878 (4) | **0.906 (1)** |
| MNIST-C-Stripe | **0.986 (1)** | 0.965 (6) | 0.977 (3) | 0.500 (9) | 0.978 (2) | 0.476 (10) | 0.969 (4) | 0.742 (8) | 0.900 (7) | 0.968 (5) |
| Shuttle | 0.988 (3) | 0.979 (7) | **0.997 (1)** | 0.980 (6) | 0.995 (2) | 0.562 (10) | 0.953 (8) | 0.754 (9) | 0.981 (5) | 0.984 (4) |
| Waveform | **0.815 (1)** | 0.742 (2) | 0.719 (5) | 0.616 (10) | 0.734 (3) | 0.715 (6) | 0.701 (7) | 0.619 (9) | 0.661 (8) | 0.722 (4) |
| Cardio | **0.945 (1)** | 0.918 (5) | 0.920 (4) | 0.872 (8) | 0.927 (2) | 0.788 (9) | 0.923 (3) | 0.597 (10) | 0.879 (7) | 0.898 (6) |
| AD-Bottle | 0.966 (2) | 0.936 (9) | 0.949 (7) | 0.500 (10) | 0.951 (5) | 0.960 (4) | 0.951 (5) | 0.939 (8) | **0.977 (1)** | 0.966 (2) |
| Census | **0.638 (1)** | 0.478 (10) | 0.609 (4) | 0.616 (3) | **0.638 (1)** | 0.538 (8) | 0.529 (9) | 0.555 (7) | 0.609 (4) | 0.587 (6) |
| Wine | 0.883 (3) | 0.796 (6) | 0.753 (9) | 0.756 (8) | 0.777 (7) | **0.917 (1)** | 0.889 (2) | 0.806 (5) | 0.395 (10) | 0.835 (4) |
| Musk | **1.000 (1)** | **1.000 (1)** | **1.000 (1)** | **1.000 (1)** | **1.000 (1)** | 0.430 (10) | 0.986 (7) | 0.783 (8) | 0.706 (9) | **1.000 (1)** |
| AVG Rank | 1.84 | 5.68 | 4.63 | 7.26 | 3.63 | 6.63 | 5.16 | 7.00 | 6.63 | 4.68 |
| p-value | NA | 0.00594816 (+) | 0.00350795 (+) | 0.00149864 (+) | 0.00543370 (+) | 0.00543370 (+) | 0.00704854 (+) | 0.00100708 (+) | 0.00065231 (+) | 0.00704854 (+) |

**Results and discussion.** Tables 11 show that ADERH consistently outperforms the other methods under their best-tuned configurations, achieving an average rank of 1.84 in AUC-ROC. Comparing these optimally tuned results to the default-parameter results (Table 1) shows that ADERH's performance advantage remains robust: even when the other isolation-based methods (IForest, EIF, PIDForest, INNE) and non-isolation methods (LOF, LODA, DeepSVDD, SLAD) are fully tuned, they generally do not match ADERH's detection accuracy.

The key to ADERH's strong performance lies in its novel design:

- *Random pairing* of points in each subset,

- *Halving* the pairwise distance to form two *compact* hyperspheres (rather than one large one),

- and a combined distance- ($\mathrm{Pitch}$) and density-based ($\mathrm{NDensity}$) scoring mechanism.

Table 12: AUC-PR performance of Multiplicative $\big((1 - \text{NDensity}) \times \text{Pitch}\big)$ vs. Additive $\big((1 - \text{NDensity}) + \text{Pitch}\big)$

| Dataset | Multiplicative | Additive |
|---|---|---|
| Optdigits | 0.061 (2) | **0.094 (1)** |
| Wbc | **1.000 (1)** | 0.080 (2) |
| Lymphography | **1.000 (1)** | 0.144 (2) |
| Celeba | **0.060 (1)** | 0.027 (2) |
| Skin | 0.345 (2) | **0.346 (1)** |
| Pendigits | **0.309 (1)** | 0.140 (2) |
| Wdbc | **0.614 (1)** | 0.168 (2) |
| AD-Toothbrush | **0.840 (1)** | 0.785 (2) |
| Wpbc | 0.261 (2) | **0.281 (1)** |
| AD-Leather | **0.975 (1)** | 0.885 (2) |
| Satimage-2 | **0.957 (1)** | 0.124 (2) |
| Backdoor | **0.222 (1)** | 0.131 (2) |
| MNIST-C-Stripe | **0.699 (1)** | 0.104 (2) |
| Waveform | 0.144 (2) | **0.235 (1)** |
| Cardio | **0.588 (1)** | 0.366 (2) |
| AD-Bottle | **0.940 (1)** | 0.841 (2) |
| Census | 0.075 (2) | **0.094 (1)** |
| Wine | **0.262 (1)** | 0.226 (2) |
| Musk | **1.000 (1)** | 0.996 (2) |
| AVG Rank | 1.26 | 1.74 |

This approach reduces hypersphere overlap with anomalies, preserves coverage of normal clusters, and robustly distinguishes boundary anomalies. The results in Tables 11 further confirm that even with grid-searched hyperparameters, the other isolation-based and non-isolation-based methods do not replicate these advantages.

## R    Ablation study: multiplicative vs. additive fusion

ADERH's anomaly scoring function (Definition 3.9) incorporates two core signals: (1) $\text{Pitch}(x, \mathcal{H})$, a ratio-based distance metric that increases for boundary points, and (2) $\text{NDensity}(\mathcal{H})$, which measures how densely populated the hypersphere is. Multiplying these signals as

$$\big(1 - \text{NDensity}(\mathcal{H})\big) \times \text{Pitch}(x, \mathcal{H})$$

ensures that a high final score occurs only when both the boundary cue ($\text{Pitch} \approx 1$) and the sparsity cue ($1 - \text{NDensity} \approx 1$) are simultaneously strong. In contrast, an *additive* combination $\big((1 - \text{NDensity}) + \text{Pitch}\big)$ may inflate scores even when only one signal is large (e.g., if $\text{Pitch} \approx 1$ but $\text{NDensity} \approx 1$ in a dense, likely normal region). By multiplying, contradictions are naturally suppressed, and each factor remains dimensionless in $[0, 1]$, so their product also comfortably stays in the interval $[0, 1]$ without extra calibration.

**Empirical findings.**    To validate this design choice, we performed an ablation study comparing the above *multiplicative* variant with its *additive* counterpart, under the same experimental pipeline. Table 12 demonstrates that across most datasets, the multiplicative scheme $\big((1 - \text{NDensity}) \times \text{Pitch}\big)$ achieves higher or comparable detection performance. While the additive combination occasionally shows slight improvements in a few datasets, the average rank metric (1.26 vs. 1.74) clearly favors the multiplicative approach overall. These findings confirm that blending boundary and density cues *multiplicatively* is better at suppressing anomalies in dense regions while still highlighting borderline outliers. Therefore, we adopt the multiplicative form as the default scoring mechanism in our anomaly detection framework.

## S   Limitations

While ADERH mitigates distance concentration through local hypersphere pairing, its effectiveness still depends on the stability of distance structures in high-dimensional spaces. When intrinsic dimensionality is large, local neighborhoods become less informative, reducing the discriminative power of geometric cues. This limitation highlights the need for feature transformations that preserve local contrast and enhance separability in complex data manifolds.

## T   Future work

Since the distribution of anomalies plays an important role in anomaly detection, it would be interesting to explore how deep learning could transform the feature space of the data so that anomalies are pushed even further away from normal data points in the first step Pang et al. [2021]. Subsequently, ADERH could leverage the transformed space to operate more effectively, potentially improving the accuracy of anomaly detection. Additionally, this approach could address challenges associated with high-dimensional data by mitigating the effects of the curse of dimensionality and improving the representation of underlying patterns in complex datasets.

