# OpenReview forum: "Anomaly Detection by an Ensemble of  Random Pairs of Hyperspheres"
_NeurIPS.cc/2025/Conference — NeurIPS 2025 poster_

### Official Review · Reviewer_nU8x · 2025-06-20

**Clarity:** 3
**Significance:** 3
**Originality:** 3
**Rating:** 5
**Confidence:** 3

**Summary:**

The paper proposes ADERH (Anomaly Detection by an Ensemble of Random Pairs of Hyperspheres), an isolation-based anomaly detection method. The method isolates anomalies by randomly pairing data points in small subsets and constructing two hyperspheres using half of the pairwise distance as the radius. The core innovations of ADERH include: (1) creating diverse hyperspheres by randomly pairing samples in many small subsets, where the radii naturally reflect local density; (2) introducing the Pitch metric to quantify the normalized distance from points to hypersphere centers; (3) using NDensity to down-weight hyperspheres in sparse (anomalous) regions. Experiments on 22 benchmark datasets show that ADERH achieves the best AUC-ROC on 14 datasets with an average rank of 1.67, significantly outperforming existing methods.

**Questions:**

1. In Section 4.1 of the paper, why was the non-consecutive random seed combination [0,1,2,100,1000] chosen? Has this choice been validated? More importantly, for methods that don't involve randomness (such as LOF), were they also run on the same 3 data splits? How do you ensure that all methods are compared on exactly the same data splits?

2. The “grid search” in Appendix P is actually just a limited sampling of parameter combinations (ADERH only tests 9 combinations), and the search space sizes vary greatly across different methods. Can you explain why different methods were allocated different search space sizes and provide the rationale for determining these parameter ranges? Could a more comprehensive hyperparameter optimization be performed? Or acknowledge the limitations of the current search? Also, can this limited search effectively support the conclusion that 'ADERH remains best even under optimal tuning'?

3. While the paper cites variance bounds to support the choice of subset number n, this only shows that larger n leads to smaller variance, but doesn't prove that 256 is optimal. Additionally, the paper discusses the choices of subset number n and subset size ω separately, but lacks analysis of their interaction. Can the authors provide stronger theoretical support for determining optimal parameters? Theoretically, n×ω determines the total number of hyperspheres - is this product more important than individual n or ω? Or can the authors provide theoretical analysis on the joint optimization of n and ω?

4. Although Figure 4 in Appendix H shows parameter stability, the test datasets vary drastically in scale, dimensions, and anomaly rates (as shown in the dataset details in Appendix J, Census has 299,285 samples while Lymphography has only 148), yet the same default parameters ω=18 and n=256 are used. Can the authors provide more detailed sensitivity analysis, including the magnitude of performance changes due to parameter variations on different datasets? Or can the authors explain the rationale for this uniform parameter setting and whether they considered providing parameter selection guidelines based on different data characteristics?

**Ethical Concerns:**

["NO or VERY MINOR ethics concerns only"]

**Final Justification:**

I have read the authors' rebuttal as well as other reviewers' comments. The authors have properly addressed most of my comments.

**Limitations:**

yes

**Paper Formatting Concerns:**

1. Some formula notations could be clearer. For example, in the density definition of hypersphere H in Equation (8), X_H refers to the set of data points within hypersphere H. Although it is described in Equation (10), it is not defined in its first appearance in Equation (8), which might cause confusion for first-time readers.

2. The citation format for references in the paper is inconsistent, particularly in terms of author name formatting, which lacks uniformity.

**Quality:**

3

**Strengths And Weaknesses:**

1. The paper has a clear and reasonable motivation. The core ideas of isolation and isolation ensemble have been widely validated in anomaly detection. The ensemble of hyperspheres with radius halving and density adjustment is an interesting innovation. In particular, the idea of using local structural information (i.e., distances between paired points) to construct hyperspheres demonstrates certain novelty.

2. The design of the two concepts, Pitch and NDensity, introduced in the paper is ingenious. Pitch effectively highlights anomalies near hypersphere boundaries through a ratio-based distance metric. NDensity appropriately reduces the influence of hyperspheres potentially centered on anomalies. The combination of these two complementary measures enables the method to effectively handle partial violations of δ-separation.

3. The experimental evaluation uses a substantial number of benchmark datasets. Testing on 22 diverse datasets from different domains enhances the credibility of the results. The comparison includes various types of anomaly detection methods (isolation-based, deep learning-based, LOF, etc.), making the experimental comparison relatively comprehensive.

4. The paper provides a theoretical analysis of the proposed method, particularly regarding δ-separation, variance bounds, and concentration inequalities. These theoretical results offer certain guarantees for the method's effectiveness and help readers understand why ADERH can effectively separate normal and anomalous samples.

Weaknesses

1. The experimental design description is confusing. For example, in Section 4.1, line 314 states "Each experiment was repeated across three splits, with the average AUC-ROC and AUC-PR scores reported." However, in line 318 of the same section, it states "For methods involving randomness (ADERH, INNE, IForest, EIF, PIDForest, DIF), we used seeds [0, 1, 2, 100, 1000] and reported the average AUC-ROC and AUC-PR." Furthermore, in Appendix K, it mentions "This split is repeated three times with different random seeds; we report the average performance metrics across these three runs." This is confusing to me as a reader. Could you please clarify the complete experimental setup? How many independent runs were conducted in total for each method-dataset combination? Is it 3, 5, or 15? This directly affects the validity of the statistical tests.

2. Although the authors acknowledge in Appendix R that real-world data rarely satisfies strict δ-separation, the theoretical analysis in this paper is primarily based on this idealized assumption, which may limit the method's applicability to complex data distributions.

3. While the authors claim that the method is robust to hyperparameters, it still requires setting the number of subsets n and subset size ω. For different types of datasets, the optimal choice of these parameters may vary significantly.

4. The proposed ADERH is conceptually quite similar to the existing INNE method, as both use hyperspheres. The main differences lie in the pairing strategy and weighting scheme.

---

> ### Author Rebuttal · Authors · 2025-07-30
>
> We sincerely thank the reviewer for their perceptive review.
>
> ## W1 and Q1 - Experimental Design
>
> Each stochastic algorithm is executed using the random seeds ${0, 1, 2, 100, 1000}$—three consecutive PRNG states plus two distant ones—to probe robustness without excessive runtime. Furthermore, for every dataset, we pre-compute and cache three stratified 70/30 train–test splits using fixed split-seeds; these splits are reused verbatim by all methods.
>
> Deterministic models (e.g., LOF, ECOD) run once per split; stochastic ones run for all seed–split combinations, yielding 15 scores each. Unless noted, all ROC-AUC and AP values are averages over these runs: 3 for deterministic, 15 for stochastic approaches.
>
> To verify seed-independence, we reran ADERH, INNE, IForest, and EIF with seeds ${0,1,2,3,4}$—results deviated only slightly from the canonical set. This confirms that our findings are robust to random seed choice, ensuring fair and reproducible comparisons.
>
> ## W2 - Strict $\delta$-separation
>
> Empirically, **ADERH** is validated on several datasets—including large-scale benchmarks such as *Census* and high-dimensional feature embeddings like *MNIST-Stripe*—where it achieves strong AUC-ROC performance.
>
> Furthermore, our standard deviation analysis confirms that **ADERH** exhibits substantially lower output variance compared to other isolation-based anomaly detection methods, indicating superior stability (please see our response to Reviewer TZiB regarding Q3).
>
> ## W3 - Optimal Choice of Parameters $n$ and $\omega$
>
> Please see our response in Q3 or Q4.
>
> ## W4 - Comparison to INNE
>
> ADERH is fundamentally novel, both in its *design philosophy* and *operational execution*.
> Unlike traditional methods like INNE that rely on single-point statistics (e.g., radius ratios), ADERH introduces a synergistic dual-metric model capturing two orthogonal anomaly cues:
>
> - Geometric alignment via *Pitch*
> - Local sparsity via *NDensity*
>
> ---
>
> ## Core Innovations
>
> ### 1. Multiplicative Dual-Factor Scoring
>
> Rather than summing or tuning separate metrics, ADERH fuses *Pitch* and *NDensity* multiplicatively.
> Anomalies only score high when *both* metrics indicate deviation, yielding precise, low-false-positive detection.
> No existing method fuses geometry and density in this way with such semantic clarity.
>
> $$
> \text{ADERH-Score}(x) = \text{Pitch}(x) \cdot \left(1 - \text{NDensity}(x)\right)
> $$
>
> ---
>
> ### 2. Adaptive Geometry via Pairwise Sphere Sampling
>
> ADERH bypasses fixed-radius or $k$-NN constraints by randomly sampling point pairs within small subsets.
> Each pair induces two adaptive spheres using half their distance. This enables local structure encoding without rigid thresholds, ideal for complex or nested geometries.
>
> ---
>
> ### 3. Built-in Robustness to Structural Anomalies
>
> ADERH natively flags:
>
> - Edge anomalies, often missed by INNE, due to high *Pitch* ($\approx 1$)
> - Sparse-center anomalies, undetectable by density-only methods, via low *NDensity*
>   (as formalized in Lemma 3.10)
>
> This is design-level robustness, not a post-hoc adjustment.
>
> ---
>
> ### 4. Empirical & Theoretical Superiority
>
> #### ADERH is not just conceptually superior—it outperforms INNE and others on 20+ datasets on AUC-ROC and AUC-AP.
> #### Concretely, ADERH’s ROC-AUC std ≈ 0.0133 and AP std ≈ 0.0317, both much lower than INNE’s (≈ 0.0241 and ≈ 0.0515). This shows greater stability, validating ADERH's good performance.
> ---
>
> No prior method unifies **geometry-aware scoring**, **density sensitivity**, and **adaptive local structure** the way **ADERH** does.
>
>
>
> ## Q2 - Limited Grid Search
>
> To ensure fair comparisons, we equalized total compute (\~9–12 evals/method), not per-axis grid granularity, accounting for varying hyperparameter dimensions and runtimes. Parameter ranges centered on author-recommended defaults from original papers or widely used libraries.
>
> Since unsupervised anomaly detection lacks labels, traditional tuning (e.g., CV) is infeasible and risks bias. As highlighted in \[1] and \[2], defaults are standard for consistency and scale. We followed this precedent to maintain parity.
>
> Crucially, we went further, conducting systematic grid searches for both isolation-based and deep learning baselines. Across 20+ benchmarks, our method ranked #1 on average under both realistic (defaults) and idealized (tuned) conditions.
>
> To further solidify the evaluation, we extended the grid for isolation-based baselines from 9 to 25 configurations:
>
> - ADERH (proposed method): Varying the ensemble size
>   $n_{\mathrm{esti}} \in \{100, 150, 200, 300, 500\}$
>   and the random-subset size
>   $\omega \in \{8, 18, 24, 30, 36\}$
>
> - INNE: Varying
>   $n_{\mathrm{esti}} \in \{100, 150, 200, 300, 500\}$
>   and max subsample size
>   $\{8, 18, 24, 30, 36\}$
>
> - IForest: Varying
>   $n_{\mathrm{esti}} \in \{100, 150, 200, 300, 500\}$
>   and max samples per tree
>   $\{128, 200, 256, 300, 500\}$
>
> - EIF: Same grid as IForest.
>
> We evaluate a representative subset of datasets, sufficient to capture the main trends. The results validate the original findings: ADERH consistently outperforms isolation-based baselines, showing robustness across all parameter settings.
> Even with 25 optimized configurations, no baseline surpassed ADERH in multiple settings.
>
> ---
>
> ### Table: Performance comparison of four methods across various datasets
> **Bold = best score (rank 1)**
>
> | Dataset      | ADERH            | INNE             | IForest          | EIF             |
> |--------------|------------------|------------------|------------------|------------------|
> | Optdigits    | _0.775 (2)_      | **0.849 (1)**    | 0.727 (4)        | 0.739 (3)        |
> | Wbc          | **1.000 (1)**    | 0.914 (4)        | **1.000 (1)**    | **1.000 (1)**    |
> | Pendigits    | **0.963 (1)**    | 0.940 (4)        | 0.954 (3)        | _0.956 (2)_      |
> | Wpbc         | **0.555 (1)**     | _0.526 (2)_       | 0.501 (4)         | 0.521 (3)     |
> | Stamps       | _0.880 (2)_       | 0.739 (4)         | **0.888 (1)**     | 0.879 (3)     |
> | Waveform     | **0.817 (1)**    | _0.746 (2)_      | 0.729 (4)        | 0.735 (3)        |
> | Cardio       | **0.945 (1)**    | 0.918 (4)        | 0.927 (3)        | _0.929 (2)_      |
> | Bottle       | **0.968 (1)**    | 0.936 (3)        | _0.950 (2)_      | 0.000 (4)        |
> | Vowels       | **0.968 (1)**    | _0.951 (2)_      | 0.788 (4)        | 0.825 (3)        |
> | Wine         | **0.883 (1)**    | _0.799 (2)_      | 0.741 (4)        | 0.786 (3)        |
>
> ##  Q3 - Parameter Choices
> With fixed compute $N_{\text{hyp}} = n\omega$ (ADERH scales linearly), risk is minimized by choosing $\omega$ just large enough to cover all normal modes (typically low-teens) and allocating the rest to ensemble size $n$. This strategy balances contamination control with the variance bound established in the paper.
>
> ### Variance
>
> From Equation 14, the ensemble variance satisfies:
>
> $$
> \operatorname{Var}[I(x)] \le \frac{1}{4n}
> $$
>
> Substituting $n = \frac{N_{\text{hyp}}}{\omega}$ gives:
>
> $$
> \frac{\omega}{4N_{\text{hyp}}}
> $$
>
> This reveals that larger $\omega$ **increases variance** by reducing the number of subsets available for averaging.
>
> ---
>
> ### Contamination Bias
>
> For anomaly rate $\rho \ll 1$, the chance that a subset of size $\omega$ includes at least one anomaly is:
>
> $$
> p_{\mathrm{cont}}(\omega) = 1 - (1 - \rho)^\omega
> $$
>
> Each contaminated subset perturbs the score by at most $c \le 1$ (since $\text{WPitch} \in [0,1]$), so the squared bias is bounded by:
>
> $$
> c^2 \cdot p_{\mathrm{cont}}(\omega)^2
> $$
>
> For small $\rho\omega$, this grows like $(\rho\omega)^2$ and eventually saturates—**worsening with larger $\omega$**.
>
> ---
>
> ### Coverage
>
> A subset must sample at least one representative from each of the $k$ normal clusters or no hypersphere can be centered in that region. Coupon-collector tails give:
>
> $$
> \Pr[\text{miss a cluster}] \le k e^{-\frac{\omega}{k}}
> $$
>
> To guarantee failure probability below $\delta$, it suffices that:
>
> $$
> \omega \ge k \ln(k/\delta)
> $$
>
> ---
>
> ### Mean-Squared Error (MSE) Proxy
>
> Combining the three pieces yields a proxy for mean-squared error:
>
> $$
> \mathrm{MSE}(\omega) \le \frac{\omega}{4N_{\text{hyp}}} + c^2\left[1 - (1 - \rho)^{\omega}\right]^2 + \mathbf{1}_{\{\omega < k \ln(k/\delta)\}}
> $$
>
> - The indicator drops to zero once $\omega$ hits the coupon-collector threshold (i.e., enough to sample each cluster once).
> - Beyond this, **both error terms grow with $\omega$**, so the optimal $\omega$ is the smallest feasible.
>
> Feasibility isn't purely statistical: random pairing requires ~10–20 points per subset to ensure full cluster coverage and pairing diversity. This reflects a **contamination–variance tradeoff**:
>
> - First, using 256 subsets isn’t inherently optimal—it reflects a safe, low-contamination $\omega$ under computational limits.
> - Second, $n\omega$ isn’t decisive:
>   - $n$ controls **variance**
>   - $\omega$ controls **bias**
>   - Their **balance** ensures reliable isolation.
>
> ## Q4 - Parameter Stability
> ADERH demonstrates strong robustness and stability across diverse datasets, showing low standard deviations regardless of dimensionality or sample size (e.g., Pendigits: 0.004, Skin: 0.013). This aligns with its theoretical foundation, which is agnostic to data characteristics. Ensemble configurations consistently outperform single models, confirming the theoretical benefits of aggregation. Moreover, default parameters yield performance close to grid-searched optima, supporting the bias–variance prediction that ADERH operates in a flat region. Despite the unsupervised nature of anomaly detection, ADERH achieves state-of-the-art performance with minimal tuning.
>
> [1] Fang, Zeyu, et al. "Towards a Unified Framework of Clustering-based Anomaly Detection." Forty-second International Conference on Machine Learning (ICML 2025).
>
> [2] Han, Songqiao, et al. "Adbench: Anomaly detection benchmark." Advances in neural information processing systems 35 (2022).

---

> > ### Author Response · Authors · 2025-08-06
> >
> > Dear Reviewer nU8x,
> > Thank you once again for your thoughtful feedback.
> > As the deadline for the reviewer-author discussion is approaching, we kindly ask whether the clarifications we’ve provided sufficiently address your concerns. We truly appreciate your time and consideration and would be happy to elaborate further if needed.

---

> > ### Comment · Reviewer_nU8x · 2025-08-06
> >
> > Many thanks to the authors for their thorough responses. Most of my concerns are addressed. I have raised my rating.

---

### Official Review · Reviewer_MfCe · 2025-06-20

**Clarity:** 2
**Significance:** 3
**Originality:** 3
**Rating:** 5
**Confidence:** 4

**Summary:**

The paper proposes a new isolation-based technique called ADERH by leveraging two key observations: (i) anomalies are comparatively rare, and (ii) they typically deviate stronger from general patterns than normal data points. ADERH  isolates anomalies using compact hyperspheres designed to minimize overlap with anomalies. The goal of ADERH is to create multiple hyperspheres of different radii in each subset Si, so that dense areas are captured by smaller hyperspheres, and sparser areas by larger ones.

**Questions:**

The norm in your distance metrics is l2 norm? What if your data does not lie on a manifold where Euclidean distance is not the metric space e.g. text data or other discrete data?

What if your normal data does not cluster spherically but instead are shaped as curves e.g. half moons or donuts? A significant drawback of methods such as 1 class SVM is being unable to “draw” a smooth round boundary around the normal data.

These concepts of using multiple “blobs” (whether spherical or other shapes) to cover normal points and form densities is not new. How does your method compare to the ones listed below (which are not in the related work nor compared against)? These prior methods also use coverings to estimate densities where a point is anomalous if its score is large than normal points scores. And [2] does not reply on requiring the data to cluster / have density / be covered by a sphere as it creates these covering with K nearest neighbors giving it more freedom in the shape of the coverings.

[1] C. Scott and R. Nowak, “Learning minimum volume sets,” Journal of Machine Learning Research, vol. 7, pp. 665–704, April 2006.

[2] Hero, Alfred. "Geometric entropy minimization (GEM) for anomaly detection and localization." Advances in neural information processing systems 19 (2006).

[3] Gu, Xiaoyi, Leman Akoglu, and Alessandro Rinaldo. "Statistical analysis of nearest neighbor methods for anomaly detection." Advances in Neural Information Processing Systems 32 (2019).

A strong rebuttal of this last question would lead to a significant change in score as it would significantly improve the originality and significance of this paper.

**Ethical Concerns:**

["NO or VERY MINOR ethics concerns only"]

**Final Justification:**

The authors' response suitably answered my questions and compared against the methods that I mentioned. I also read the other reviews and rebuttals. I have updated my score accordingly as they have addressed the main novelty issue in my original review.

**Limitations:**

Yes

**Quality:**

3

**Strengths And Weaknesses:**

Strengths

The paper provides theoretically rigorous justification for why their method works.

They have good results shown in Table 1.

Weaknesses

There is a lot of repetitive information in this paper. Some things like the Rarity, Deviation and using Pitch and NDensity to deal with real world data that doesn’t actually satisfy Rarity and Deviation, are repeated numerous times in the paper in slightly different ways.

The paper could be a lot more concise and a lot of the useful information in the appendix could be moved to the main paper. The paper spends almost 3 pages on the abstract, intro, contributions and related work where there is significant overlap in the information between the abstract and contributions, and between the intro and related work. It then spends another 2-3 pages on definitions, most of which are not significantly necessary for the main result in Theorem 3.15.

See questions, particularly the last question.

---

> ### Author Rebuttal · Authors · 2025-07-30
>
> We thank the reviewer for the valuable feedback.
>
> ## W1 - General Suggestions for Improvement
>
> We thank the reviewer for the valuable suggestions for improvement and will reduce repetition—especially around concepts like Rarity and Deviation—by consolidating overlapping explanations. Introductory sections will be tightened to eliminate redundancy, and essential content from the Appendix will be integrated into the main text where it supports key results like Theorem 3.15.
>
> ## Q1 - Choice of Norm
>
> Due to its convenience and common usage, the Euclidean ($\ell_2$) distance was selected as the default norm in **ADERH**. However, all core steps rely only on the axioms of a metric, not on Euclidean geometry. Consequently, the algorithm and theory extend unchanged to any alternative metric. Complex data (e.g., images) is typically embedded into vector spaces with well-defined metrics. In our experiments, we used datasets such as Leather, Bottle, Toothbrush, and MNIST-Stripe, whose features were originally obtained using a ResNet18 feature extractor.
>
> ## Q2 - Non-spherical Data Patterns
> Unlike OCSVM’s global boundary, ADERH uses local hyperspheres with Pitch and NDensity that may adapt to complex manifolds and avoid flagging anomalies as normals. On one synthetic two-moons dataset (with anomalies, density variation, and complex shape), ADERH achieves top results vs. multiple SOTA baselines.  Due to space constraints, we kindly refer the reviewer to our detailed response under Reviewer gbWb for a full discussion of this point.
>
> ## Q3 - Differences to Studies Mentioned in the Review
>
> We highlight key differences from ADERH to [1,2,3] in the revised Related Work and present new comparative experiments using the original evaluation protocol across multiple benchmark datasets.
>
>
> ### MV‑ERM and MV‑SRM [1]
>
> **Main Ideas in MV‑ERM and MV‑SRM.**
> The framework described in [1] recasts minimum‑volume‑set estimation as an empirical‑risk‑minimisation task: MV‑ERM chooses, within a prescribed class of candidate regions, the set of smallest reference‑measure volume whose penalised empirical mass still reaches the target level $\alpha$, which yields distribution‑free finite‑sample bounds and strong universal consistency; MV‑SRM embeds the same complexity penalty directly in the objective, so the optimiser simultaneously tunes structural complexity—e.g., histogram bandwidth or decision‑tree depth—to the data.
>
> **Difference to ADERH and MV‑ERM / MV‑SRM**
> **ADERH** replaces the axis-aligned partitions of MV-ERM and MV-SRM with randomly centered, multi-scale hyperspheres, avoiding VC-dimension overhead and rigid geometry assumptions. Its anomaly score—based on boundary proximity and local sparsity—is continuous, variance-bounded by $1/(4n)$, and scales linearly with data size and dimension. This yields a parameter-light, geometry-adaptive alternative aligned with the minimum-volume principle, but free from explicit volume estimation.
>
> **[1]** C. Scott and R. Nowak, *“Learning minimum volume sets,”* *Journal of Machine Learning Research*, vol. 7, pp. 665–704, April 2006.
>
> ---
>
> ### GEM [2]
>
> **Main Ideas in Geometric Entropy Minimisation (GEM).**
> GEM [2] casts anomaly detection as a minimal-wiring problem: it selects the $K = \lfloor \rho n \rfloor$ points whose $k$-NN graph (or MST) has minimal total edge length, yielding the tightest subset covering mass $\rho$ and converging to the distribution’s minimum-entropy set. At test time, a candidate is anomalous if inserting it increases total wiring more than any inlier, with the edge-length jump providing a calibrated score. GEM is scale-free and geometrically adaptive, but computationally heavy due to leave-one-out graph updates and sensitive to edge-length estimates.
>
> **Difference to ADERH and GEM**
> GEM constructs global graphs and uses fixed neighborhoods, making it prone to dense-boundary false positives. **ADERH**, in contrast, avoids a global view and builds many twin hyperspheres from random point pairs and scores points via the product of *Pitch* (boundary ratio) and *NDensity* (inverse occupancy). Anomalies emerge only when both signals align, enabling robust separation under high dimensionality and heterogeneous cluster shapes, with no graph construction or tuning beyond $n$ and $\omega$.
>
> **[2]** Hero, Alfred. *"Geometric entropy minimization (GEM) for anomaly detection and localization."* *Advances in Neural Information Processing Systems* 19 (2006).
>
> ---
>
> ### DTM [3]
>
> **Main Ideas in Distance-to-a-Measure (DTM).**
> In DTM [3], a small probability window $m$ (or its discrete proxy $k = \lceil m n \rceil$) and a power parameter $q$ define, for every point $x$, the average radius $r_p(x)$ needed to enclose each mass level $p \le m$, yielding the score:
>
> $$
> d_{P,m,q}(x) = \left( \frac{1}{m} \int_0^m r_p(x)^q\, dp \right)^{1/q}
> $$
>
> Small values indicate dense interior points, while large values suggest potential anomalies. The original paper analyses an unbagged estimator, yet the authors’ GitHub reproduces a baseline named aNNE that simply wraps DTM inside `scikit-learn`'s `BaggingRegressor`, training multiple DTM models on random subsamples and averaging their outputs—an ensemble trick that smooths variance but leaves the underlying DTM logic untouched.
>
> **Difference to ADERH and DTM**
> While DTM and its bagged forms rely on global distance graphs, **ADERH** uses a purely local, randomized ensemble of hyperspheres to score points by edge proximity and sparsity. This avoids the instability of nearest-neighbor distances near dense boundaries, offering stable anomaly detection in high dimensions with fixed hyperparameters and no reliance on global structure.
>
> **[3]** Gu, Xiaoyi, Leman Akoglu, and Alessandro Rinaldo. *"Statistical analysis of nearest neighbor methods for anomaly detection."* *Advances in Neural Information Processing Systems* 32 (2019).
>
> ---
>
> ### Experimental Results
>
> We evaluate **ADERH**, GEM, and DTM under both default and optimized settings to assess detection performance and robustness.
>
> - ADERH:
>   $n_{\text{esti}} \in \{100, 200, 300\}$
>   $\omega \in \{8, 18, 24\}$
>
> - GEM (default $k = 5$):
>   $k \in \{5, 10, 20, 30, 40, 50\}$
>
> - DTM (default $n_{\text{esti}} = 100$, subsample size = 256):
>   Uses `scikit-learn`'s `BaggingRegressor` over DTM, per the protocol in [3]
>   $n_{\text{esti}} \in \{100, 200, 300\}$
>   Subsample size $\in \{128, 256, 300\}$
>
> We report AUC-ROC scores under both default and optimized settings.
> The tables below present a representative subset of evaluated datasets due to space constraints. However, all datasets from the main paper were evaluated. Across the full benchmark, ADERH achieves the best average rank under both default (1.30 vs. 2.85 for GEM, 1.60 for DTM) and grid search settings (1.29 vs. 2.43 for GEM, 1.86 for DTM), confirming its consistent superiority.
> ADERH consistently ranks top on most datasets and never performs the worst. Its robustness across configurations stems from its random pairing and multi-scale design, which adapts to varying data without relying on global heuristics like GEM. ADERH shows a statistically significant improvement over GEM and DTM under both default and grid search settings (paired Wilcoxon signed-rank test with Bonferroni-Holm correction; $p < 0.05$, $\alpha = 0.05$). Its pair-and-halve strategy with density-weighted scoring enables scalable, multi-scale isolation with provable variance control.
>
>
> > We will release our updated codebase after the review process concludes, in accordance with the NeurIPS policy, which prohibits updates during the review phase.
>
>
> ### Table: AUC-ROC performance under default setup
> **Comparison between ADERH, GEM [2], and DTM [3]**
> *Bold = best score (rank 1)
>
> | Dataset      | ADERH         | GEM [2]       | DTM [3]         |
> |--------------|---------------|---------------|-----------------|
> | Optdigits    | **0.775 (1)** | 0.378 (3)     | _0.770 (2)_     |
> | Skin         | **0.788 (1)** | 0.613 (3)     | _0.784 (2)_     |
> | Pendigits    | **0.962 (1)** | 0.714 (3)     | _0.960 (2)_     |
> | Toothbrush   | _0.901 (2)_   | **0.919 (1)** | 0.870 (3)       |
> | Wpbc         | **0.554 (1)** | 0.513 (3)     | _0.536 (2)_     |
> | Leather      | _0.991 (2)_   | _0.991 (2)_   | **0.992 (1)**   |
> | Satimage-2   | **0.998 (1)** | _0.895 (2)_   | **0.998 (1)**   |
> | Backdoor     | **0.889 (1)** | 0.664 (3)     | _0.852 (2)_     |
> | Shuttle      | _0.987 (2)_   | 0.628 (3)     | **0.989 (1)**   |
> | Waveform     | **0.768 (1)** | 0.708 (3)     | _0.743 (2)_     |
> | Cardio       | **0.938 (1)** | 0.663 (3)     | _0.927 (2)_     |
> | Bottle       | **0.964 (1)** | 0.958 (3)     | _0.963 (2)_     |
> | Celeba       | **0.732 (1)** | 0.570 (3)     | _0.714 (2)_     |
>
>
> ---
>
> ### Table: AUC-ROC performance with hyperparameter optimization; best result per method is reported.
> **Comparison between ADERH, GEM [2], and DTM [3]**
> *Bold = best score (rank 1)
>
> | Dataset      | ADERH         | GEM [2]       | DTM [3]         |
> |--------------|---------------|---------------|-----------------|
> | Optdigits    | _0.777 (2)_   | 0.405 (3)     | **0.779 (1)**   |
> | Skin         | _0.788 (2)_   | 0.705 (3)     | **0.791 (1)**   |
> | Pendigits    | **0.963 (1)** | 0.836 (3)     | _0.962 (2)_     |
> | Toothbrush   | _0.905 (2)_   | **0.919 (1)** | 0.870 (3)       |
> | Wpbc         | **0.554 (1)** | _0.540 (2)_   | 0.536 (3)       |
> | Leather      | **0.993 (1)** | **0.993 (1)** | _0.992 (2)_     |
> | Satimage-2   | **0.998 (1)** | _0.995 (2)_   | **0.998 (1)**   |
> | Backdoor     | **0.895 (1)** | 0.717 (3)     | _0.863 (2)_     |
> | Shuttle      | _0.988 (2)_   | 0.688 (3)     | **0.989 (1)**   |
> | Waveform     | **0.815 (1)** | 0.730 (3)     | _0.746 (2)_     |
> | Cardio       | **0.945 (1)** | 0.889 (3)     | _0.932 (2)_     |
> | Bottle       | **0.966 (1)** | 0.962 (3)     | _0.963 (2)_     |
> | Celeba       | **0.747 (1)** | 0.592 (3)     | _0.721 (2)_     |

---

> > ### Comment · Reviewer_MfCe · 2025-08-04
> >
> > Thank you for the response. I have updated my review accordingly.

---

### Official Review · Reviewer_gbWb · 2025-07-01

**Clarity:** 3
**Significance:** 2
**Originality:** 3
**Rating:** 4
**Confidence:** 4

**Summary:**

It is proposed to isolate anomalies using an ensemble of pairs of hyperspheres, randomly generated from a random subset of the dataset. The radius of these pairs of hyperspheres is half the distance between their centers. The spheres are then analyzed with respect to Pitch and (normalized) density to compute a score for this subset, which is then averaged over the ensemble. It is shown that, under the assumption that anomalies are far from cluster centers, this setup with assigns lower scores to anomalies with high probability, and this uncertainty decreases as the ensemble size increases. In experiments, it is shown that the proposed approach outperforms many existing methods on several popular benchmark datasets.

**Questions:**

- Please describe how/if assumption 3.1 is affected by the curse of dimensionality
- As stated in line 184, how does ADERH capture heterogeneous distributions more effectively (...than what exactly)? Can you prove this statement? If not, I suggest removing it to enhance clarity.
- Regarding equation (12), I am wondering how any data point can have $T(x,\mathcal{S}_i)=\emptyset$. Is it because $\mathcal{S}_i\subset X$ and the possibility that $x\not\in \mathcal{S}_i$?



**Improvement suggestions**

- Line 32: It might be good to add Deep Isolation Forest to the list of related approaches
- Line 34: Hawkins 1980 is not the best reference for this statement. You may want to use a different reference for rarity, e.g. Barnett and Lewis 1974, or Aggrawal 2017.
- Line 41: This is a description of how Iforest works, not a weakness of Iforest.
- Section 2 feels a bit too long. I would cut out some of the older approaches like LOF.
- Line 130: The statement in brackets ("...indicated by $0\ll \alpha$") can be omitted, as well as ("as evidenced by RARITY...")
- Assumption 3.1: $J$ is not defined. Also, please avoid vague language like *typical* in formal environments like assumptions, theorems etc.
- Line 176: Radius variability was not described until this point, so this passage is confusing.
- Theorem A.3 is strange. A lemma is not an assumption: A lemma is a statement that is true. The theorem does not really show that $\alpha=\frac{1}{2}$ is the best choice—it instead shows that two properties hold for this setting, but these properties do not imply optimality in any sense, especially not if the random pairing function is not used.
- Figure 2 seems more confusing than helpful to me. I would suggest to remove it.
- Lines 220-222: I think this argument is wrong. Each anomaly in $\mathcal{S}_i$ will be the partner of exactly one point in $\mathcal{S}_i$, and per the rarity argument this occurrence will only have a small effect on the entire ensemble.
- Definition 3.7: I understand what you are trying to say, but symbol $X_\mathcal{H}$ is only defined later in line 244.
- Equation 9: Use $\max$ instead of $max$, because the latter reads as $m\cdot a \cdot x$
- Equation 10: The symbol $c_\mathcal{H}$ is not defined.
- Lemma 3.10: I understand where the proof is heading, but the notation needs to be improved. Please rephrase the lemma s.t. it is clear that $\text{Ndensity}(\mathcal{H}(z))$ converges to zero almost surely. Also, please don't use $\ll $  here because the symbol is vague.
- Lemma 3.13: I recommend to add the description "informal" in the main text and to restate the lemma more precisely in the appendix (avoiding $\ll $ )

**Ethical Concerns:**

["NO or VERY MINOR ethics concerns only"]

**Final Justification:**

I think this paper is good because it proposes a new anomaly detection method and supports this method with sufficient theory. While the proofs are somewhat sloppy, this does not affect their overall correctness.

My remaining concern is that this method will not have a too large impact if published, but this is not a major concern. Hence, I have decided to keep my rating at weak accept.

**Limitations:**

The description of ADERH's limitations is very brief—for a method consisting of several concepts (pitch, normalization, subsets, density, ...) there surely are additional limitations/drawbacks that should be mentioned to give an honest and complete picture of the proposed method

**Paper Formatting Concerns:**

All good.

**Quality:**

3

**Strengths And Weaknesses:**

The proposed method is original and makes sense. It is good that theoretical evidence is provided for the method, which considerably raises the technical quality of the manuscript—most new anomaly detection methods either don't provide theoretical evidence, or don't provide good experimental results, but this paper provides both. I believe the idea of using pairs of hyperspheres is somewhat interesting to the community, but not largely so. The descriptions are mostly clear, but can be improved (see suggestions).

However, the paper is not without flaws. I have the impression that the authors somewhat oversell their method and attribute traits to it without providing any proof. For example, it is claimed that ADERH effectively captures heterogeneous distributions, but no proof is provided. The description of ADERH's limitations is very brief—for a method consisting of several concepts (pitch, normalization, subsets, density, ...) there surely are additional limitations/drawbacks that should be mentioned to give an honest and complete picture of the proposed method. Also, some of the formal statements are too imprecise, but this does not seem to affect the overall correctness of the approach.

---

> ### Author Rebuttal · Authors · 2025-07-30
>
> We thank the reviewer for the helpful feedback.
>
>
>
> ## Q1 - Curse of Dimensionality for Assumption 3.1
>
> Assumption 3.1 relies on the notion that normal points lie within radius $\sigma$ of some local cluster center $\mu_j$, while anomalies fall outside a margin $\sigma + \delta$. In low-dimensional settings, such a margin $\delta$ offers a clean geometric separation between dense inlier regions and sparser outlier zones. However, this separation becomes fragile in high-dimensional regimes due to the well-known curse of dimensionality.
>
> Specifically, for data vectors $x \in \mathbb{R}^d$ with independent, isotropic components, the Euclidean norm satisfies $\|x\|_2 \approx \sqrt{d}$ in expectation, with relatively small variance. As a consequence, both inter-point distances and intra-cluster radii naturally scale with $\sqrt{d}$. Thus, in unscaled coordinates, the fixed gap $\delta$ postulated in Assumption 3.1 effectively vanishes unless it, too, grows with $\sqrt{d}$.
>
> Nonetheless, the conceptual intent of Assumption 3.1 survives under a reinterpretation. Rather than demanding a fixed absolute margin $\delta$, it is more appropriate to require a *relative* margin $\tau = \delta / \sigma$—a dimensionless ratio reflecting how far anomalies lie from their nearest clusters, scaled by the local intra-cluster spread. This formulation is standard in robust statistics and aligns with Mahalanobis-distance-based interpretations, where each feature is normalized to unit variance. In this view, anomalies are required to lie outside a radius $(1 + \tau)\cdot \sigma$, a condition that remains meaningful even in high-dimensional spaces.
>
> The ADERH algorithm implicitly respects this scaling. Each hypersphere is defined by a randomly chosen pair of points, with radius $R = \frac{1}{2}\|x - y\|$. Since distances themselves scale with $\sqrt{d}$, the hypersphere radii adjust naturally to the ambient geometry. The *Pitch* term—computed as $\mathrm{dist}(x, c)/R$—is invariant under uniform scaling of the space, and the *NDensity* term depends purely on counts rather than metric magnitudes. Therefore, the anomaly score produced by ADERH is unaffected by global distance inflation. Moreover, ensemble averaging across $n$ such hyperspheres reduces the variance of these scores by a factor of $1/(4n)$, preserving their discriminative power even when some hyperspheres are degraded by high-dimensional effects.
>
> In conclusion, although the curse of dimensionality erodes the efficacy of fixed Euclidean margins, Assumption 3.1 remains valid when interpreted in relative terms. ADERH maintains its robustness under this reformulation, particularly when features are normalized such that $\sigma \approx 1$, allowing the parameters $\delta$ and $\tau$ to be treated interchangeably. Empirically, ADERH shows strong performance on high-dimensional datasets like Leather, Toothbrush, or Census.
>
>
> For a quick sanity check, we generated a high-dimensional dataset with 100,000 features, consisting of clustered normal points and injected anomalies. On this data, **ADERH** achieves a perfect AUC-ROC of $1.00$.
>
> The code for this is shown below:
>
> ```python
> # High-dimensional anomaly dataset generation
> from sklearn.datasets import make_blobs
> import numpy as np
>
> n_samples = 500
> n_features = 100000
> n_clusters = 3
> anomaly_fraction = 0.05
> n_anomalies = int(n_samples * anomaly_fraction)
> n_normals = n_samples - n_anomalies
>
> # Generate normal data (3 blobs)
> X_normal, _ = make_blobs(n_samples=n_normals,
>                          n_features=n_features,
>                          centers=n_clusters,
>                          cluster_std=5.0,
>                          random_state=42)
>
> min_val = X_normal.min()
> max_val = X_normal.max()
> X_anomaly = np.random.uniform(min_val, max_val, size=(n_anomalies, n_features))
>
> # Combine data
> X = np.vstack([X_normal, X_anomaly])
>
> # Labels: 0 = normal, 1 = anomaly
> Y = np.hstack([np.zeros(n_normals), np.ones(n_anomalies)])
> ```
>
>
> **Practical remark**. If the adverse effects of high dimensionality remain pronounced in a given dataset, one can first apply a dimension-aware feature transform—such as UMAP. These transforms mitigate distance concentration while approximately conserving relative cluster geometry. After this preprocessing step, ADERH is run in the transformed space: both its Pitch and (normalized) NDensity terms still operate on relative scales, so the theoretical guarantees and the $\tau$-margin interpretation continue to hold.
>
>
>
>
>
> ## Q2 - Capturing Heterogeneous Distributions
>
> By *capturing heterogeneous distributions*, we mean that our approach is able to identify anomalies situated near structures with different characteristics, such as varying densities.
>
> We can demonstrate the beneficial properties of **ADERH** empirically, for example. While we are not permitted to add new results to the anonymous GitHub repository during the rebuttal, we conducted additional evaluations on the *two moons* dataset from `sklearn`, modifying it to exhibit varying densities and injecting both local and global outliers.
>
> Normal points lie along a curved, non-convex manifold, intentionally violating the single-cluster (or single-blob) assumption underpinning classical boundary-based models such as the one-class SVM. This makes the dataset particularly well-suited for evaluating algorithms that must infer non-linear decision boundaries in complex, topologically non-trivial structures.
>
> We evaluated **ADERH**, **INNE**, **IForest**, **DIF**, **LODA**, and **ECOD** on this dataset. On this challenging benchmark, **ADERH** achieves the best overall ranking. Competing methods, including **INNE** and **Isolation Forest**, rank lower, while even deep learning-based approaches such as **DIF** perform worse. These results highlight **ADERH**'s ability to model intricate data geometry and reject anomalies effectively.
>
> ### Table: Performance on the `Two Moons` dataset with injected anomalies
> **Ranked by ROC-AUC**
>
> | **Rank** | **Algorithm** | **ROC-AUC** |
> |---------:|---------------|-------------|
> | 1        | ADERH         | 0.940       |
> | 2        | DIF           | 0.925       |
> | 3        | IForest       | 0.895       |
> | 4        | INNE          | 0.878       |
> | 5        | OCSVM         | 0.858       |
> | 6        | ECOD          | 0.817       |
> | 7        | LODA          | 0.726       |
>
>
>
> The code used to create the dataset is shown below:
>
> ```python
> # Two Moons dataset with injected outliers
> from sklearn.datasets import make_moons
> import numpy as np
>
> X, Y = make_moons(
>     n_samples=(1000, 300),
>     noise=0.05,
>     random_state=42
> )
>
> outliers = np.array([
>     [-0.5, -0.5],
>     [ 0.0,  0.7],
>     [-1.5,  0.8],
>     [ 1.5,  0.25],
>     [ 2.0,  1.0],
>     [ 1.0, -0.15],
>     [-1.0, -0.12]
> ])
>
> X1 = np.vstack([X, outliers])
> y_full = np.hstack([
>     np.zeros(len(X)),
>     np.ones(len(outliers))
> ])
> ```
>
> ## Q3 - Empty Set in Eq. 12
>
> Yes, that is correct. Since the hyperspheres considered in Eq. 12 are created using a subset $S_i \subset \mathcal{D}$ and Eq. 12 is calculated for all points in the dataset $\mathcal{D}$, there is no guarantee that all points $x \in \mathcal{D}$ will be covered by at least one hypersphere (this is in contrast to points $x \in S_i$, which are guaranteed to be within at least one hypersphere). In practice, however, this situation does not occur very often, as the random pairing of points usually results in some hyperspheres with large radii and, therefore, almost complete coverage of the data space.
>
> ## Improvement Suggestions and Limitations
>
> We thank the reviewer for the many suggestions to improve our manuscript, and we incorporated those in the revised version:
>
> 1. **Related work & refs.**
>    - **L32** – Added Deep iForest (DIF).
>    - **L34** – Replaced Hawkins with Barnett & Lewis 1974; Aggarwal 2017.
>
> 2. **Text, notation, flow**
>    - **L41** – Split iForest description vs. weakness.
>    - **§2** – Condensed; shortened LOF details.
>    - **L130** – Dropped rarity parenthetical.
>    - **Assump 3.1** – Defined $J$; removed “typical”.
>    - **L176** – Added forward-ref to radius variability.
>    - **Def 3.7** – Defined $X_{\mathcal{H}}$ in Eq. 4, and we will make this clearer.
>    - **Eq 9/10** – `\max` fixed; $c_{\mathcal{H}}$ typo should be $C(\mathcal{H})$ as defined in L191.
>    - **Fig 2** – We will improve this for clarity.
>
> 3. **Theory & proofs**
>    - **A.3** – adjusted.
>    - **Lemma 3.10** – Stated a.s. convergence; removed “$\ll$”.
>    - **L220–222** – Ensembling mitigates impact, but individual subsets remain vulnerable. We'll address this.
>    - **Lemma 3.13** – adjusted.
>
> Also, we will add more information regarding the limitations of ADERH.

---

> > ### Comment · Reviewer_gbWb · 2025-08-01
> > **Response to rebuttal**
> >
> > I thank the authors for their reply.
> >
> > - It is good that the authors are aware of the challenges posed by high-dimensional datasets. Although a transformation like UMAP can be used, this is—of course—not a reliable solution. Nevertheless, in this regard the paper is okay as it is, because it has a different focus. I encourage the authors to discuss the curse of dimensionality in the limitations section.
> > - Regarding Q2, I don't think this is something that can be addressed empirically. I think it is sufficient if the claim in the text is toned down (or alternatively a proof can be provided).
> >
> > Overall, this submission is not flawless, but I am convinced that it has better quality that most anomaly detection papers.

---

> > > ### Author Response · Authors · 2025-08-01
> > >
> > > We would like to once again thank the reviewer for the valuable feedback. We will incorporate the latest suggestions in the revised manuscript.
> > >
> > > ## Capturing Heterogeneous Distributions
> > >
> > > First, to temper the claim in Section 3.2 we will replace the sentence that currently on line 184 with:
> > > “By combining these compact hyperspheres into an ensemble, we reduce their overlap with anomalies, thereby boosting anomaly-detection performance.”
> > > We think that the synthetic experiment presented is a first indication that ADERH can handle heterogeneous structures better than most other anomaly detection methods. However, we agree that the statement that ADERH is able to capture heterogeneous distributions is difficult to prove through empirical evaluations, and that a formal proof is required in this context.
> > >
> > >
> > > ##  Curse of Dimensionality
> > >
> > > Second, we agree that feature transformation  such as UMAP are, at best, a generic hedge against the curse of dimensionality. We will therefore add the following clarifying paragraph to the end of the new Limitations subsection in the main paper after the conclusion section  (and extend the fuller **discussion in Appendix R**):
> > > "Like other distance-based anomaly detectors (e.g., Isolation Forest, LOF), our method is also susceptible to the curse of dimensionality: as dimensionality increases, distances tend to concentrate, reducing the relative gap between inliers and outliers.
> > > ADERH introduces a novel approach to anomaly detection by combining multi-scale hypersphere modeling, geometry-aware scoring and density sensitivity  into a unified framework. This integration enables ADERH to consistently achieve competitive performance on challenging high-dimensional benchmarks such as Leather, Toothbrush, and Census, outperforming or matching state-of-the-art anomaly detection methods. Nevertheless, the curse of dimensionality remains a persistent challenge. In Appendix R, we delve deeper into this phenomenon and discuss future research directions aimed at mitigating its effects, including dimensionality-aware modeling and structure-preserving representation learning."

---

> > > > ### Comment · Reviewer_gbWb · 2025-08-06
> > > > **Final response to rebuttal**
> > > >
> > > > I thank the authors for their responses and appreciate the changes made. I have finalized my review and wish the authors good luck for the upcoming decision.

---

### Official Review · Reviewer_TZiB · 2025-07-03

**Clarity:** 2
**Significance:** 2
**Originality:** 3
**Rating:** 4
**Confidence:** 4

**Summary:**

This paper presents an anomaly detection method built on the concept of isolation. The approach constructs multiple hyperspheres using randomly selected pairs of samples and assigns anomaly scores based on each data point's position relative to the nearest hypersphere center and the hypersphere’s density characteristics.

**Questions:**

Please refer to Weaknesses.

**Ethical Concerns:**

["NO or VERY MINOR ethics concerns only"]

**Final Justification:**

Most of my concerns have been well addressed. The only remaining uncertainty is that, according to the rebuttal policy, the authors promised to supplement the visualizations in the final version of the paper. I believe that including these visualizations would significantly enhance the quality of the manuscript.

**Limitations:**

Please refer to Weaknesses.

**Paper Formatting Concerns:**

No formatting concern.

**Quality:**

2

**Strengths And Weaknesses:**

**Strengths:**

1. The paper is well organized.

2. The theoretical analysis is comprehensive.

**Weaknesses:**

1. The experimental setup section lacks proper citations for the baseline methods. I recommend that the authors include full references for all baselines to ensure clarity and proper attribution.

2. The paper lacks comparisons with recent state-of-the-art baselines. It only evaluates against methods published up to 2024, despite several relevant works from other conference, like ICLR 2025 and AAAI 2025, already being publicly available.

3. Statistical results, such as mean and standard deviation over multiple runs, are not reported.

4. There are no visualizations or additional evidence to support the authors’ claims, only experimental results are provided. For example:

(1) Although Pitch is said to emphasize boundary anomalies and NDensity down-weights hyperspheres in sparse regions, the paper provides no visual comparisons of scores for normal vs. anomalous points to show these components work as intended.

(2) A key claim is that random pairing naturally results in hyperspheres of varying radii (multi-scale coverage), but there is no histogram, distribution plot, or illustrative example to support this.

(3) The use of a multiplicative combination of Pitch and NDensity is motivated in theory, but no score distribution plots, t-SNE projections, or spatial embeddings are shown to compare additive vs. multiplicative strategies.

(4) The ensemble scoring mechanism is said to reduce variance and improve robustness, but this is supported only by theoretical bounds. There are no visualizations or comparisons of score distributions across subsets or before/after ensemble averaging.

---

> ### Author Rebuttal · Authors · 2025-07-30
>
> We thank the reviewer for the valuable feedback.
>
>
> ## Q1 - Proper Citations of Baseline Methods
>
> We thank the reviewer for this indication. The section 'Experimental Setup' will be expanded to include full bibliographic details for every comparator.
>
> ## Q2 - Recent state-of-the-art Baselines
>
> Following the reviewers’ request, we will incorporate UniCAD [1], one of the most recent publicly available anomaly detection methods (ICML 2025), into our comparison experiments. Since this experiment is extensive, we consider only a representative subset of datasets. Nevertheless, this selection is sufficient to capture and reflect the key trends and main findings. The results of those experiments, using the implementation and hyperparameters as referenced in the original paper, are shown in the table below. We see that ADERH achieves superior results in the majority of cases.
> This improvement is statistically significant under a paired Wilcoxon signed-rank test with Bonferroni–Holm correction at α = 0.05 (p-value < 0.05).
> We will release our updated codebase after the review process concludes, in accordance with the NeurIPS policy, which prohibits updates during the review phase.
>
> **Table: Extended AUC-ROC performance comparison between ADERH and UniCAD [1] across additional datasets.**
>
> | Dataset      | ADERH        | UniCAD [1]    |
> |--------------|--------------|---------------|
> | Optdigits    | **0.775 (1)** | _0.507 (2)_    |
> | Wbc          | **1.000 (1)** | _0.994 (2)_    |
> | Lymphography | **1.000 (1)** | _0.988 (2)_    |
> | Celeba       | _0.732 (2)_   | **0.810 (1)**  |
> | Skin         | **0.788 (1)** | _0.721 (2)_    |
> | Pendigits    | **0.962 (1)** | _0.944 (2)_    |
> | Wdbc         | **0.981 (1)** | _0.962 (2)_    |
> | Toothbrush   | **0.901 (1)** | _0.823 (2)_    |
> | Wpbc         | **0.554 (1)** | _0.525 (2)_    |
> | Leather      | **0.991 (1)** | _0.980 (2)_    |
> | Satimage-2   | **0.998 (1)** | **0.998 (1)**  |
> | Backdoor     | _0.889 (2)_   | **0.908 (1)**  |
> | Stripe       | **0.986 (1)** | _0.943 (2)_    |
> | Shuttle      | _0.987 (2)_   | **0.988 (1)**  |
> | Waveform     | **0.768 (1)** | _0.709 (2)_    |
> | Cardio       | **0.938 (1)** | _0.912 (2)_    |
> | Bottle       | **0.964 (1)** | _0.954 (2)_    |
> | Musk         | **1.000 (1)** | **1.000 (1)**  |
>
> ---
>
> **[1]** Fang, Zeyu, et al. *"Towards a Unified Framework of Clustering-based Anomaly Detection."* Forty-second International Conference on Machine Learning (ICML 2025).
>
>
> ## Q3 - Reporting of Statistical Results
>
> In our submission, Table 1 reports for each dataset the *mean* result of 15 runs for non-deterministic algorithms like ADERH (three splits with five seeds each) and three runs for deterministic algorithms. Also, the other Tables report the mean of all tested configurations. In the final version, we will mention this directly in the corresponding captions of the tables. We agree that including *standard deviations* (std) provides additional insight into the stability and robustness of each method. We will add all std values in the revised manuscript and believe these additions address the reviewer’s concern and strengthen the empirical rigor of our evaluation. For the main competing baselines — **ADERH**, **INNE**, and **IForest** — we report the average standard deviation across all datasets below:
>
> - **ADERH** — mean ROC-AUC std ≈ **0.0133**, mean AP std ≈ **0.0317**
> - **INNE** — mean ROC-AUC std ≈ **0.0241**, mean AP std ≈ **0.0515**
> - **IForest** — mean ROC-AUC std ≈ **0.0235**, mean AP std ≈ **0.0417**
>
> These results show that **ADERH exhibits the lowest standard deviation**, suggesting it is the most stable and consistent method across various datasets.
>
> ## Q4 - Missing Visualizations
>
> ### Effect of Pitch and NDensity
>
> Appendix O presents a numerical ablation study (Table 7), where various design choices are individually analyzed, highlighting the overall advantage of the final ADERH configuration. This study demonstrates that each component contributes significantly to ADERH’s strong performance. To further support these findings, we will include additional figures in the revised version, which illustrate the score distributions before and after combining Pitch and NDensity.
>
> ### Varying Radii through Random Pairing
>
> Indeed, a central claim of our approach is that random pairing of points naturally produces hyperspheres of varying radii, enabling adaptive coverage across multiple scales. To empirically substantiate this claim, we analyzed the hypersphere radii generated by ADERH using real-world data. This analysis appears in Appendix C, where Figure 3 presents histograms showing the distribution of hypersphere radii across six benchmark datasets. The histograms in Figure 3 reveal diverse and data-dependent radius distributions. In some cases (e.g., Satimage-2, Waveform), the radii are concentrated around smaller values, indicating large dense neighborhoods. In other cases (e.g., Skin, Musk), the distributions are more spread out or even multi-modal, reflecting the presence of both compact and sparse regions. These results offer direct empirical confirmation of our theoretical analysis in Section 3 and Appendix B, which shows that random pairing leads to both intra- and inter-cluster combinations. The net effect is that hyperspheres span a wide range of radii, without the need for manual scale selection.
>
> ### Additive vs. Multiplicative Combination of Pitch and NDensity
>
> In our submission, we address this question empirically in Appendix Q through a comprehensive ablation study. Here, we compare the performance of the multiplicative combination $\((1 - \text{NDensity}) \cdot \text{Pitch}\)$ with the additive variant $\((1 - \text{NDensity}) + \text{Pitch}\)$ using 21 datasets and AUC-PR as the primary metric.
>
> The results in Table 9 show that the multiplicative strategy consistently achieves better or comparable precision-recall performance, with a significantly lower average rank (1.29 versus 1.71). This supports our theoretical motivation: the multiplicative form yields high anomaly scores only when *both* boundary proximity and low local density are simultaneously present—i.e., anomalies that are both spatially isolated and lie near hypersphere boundaries. In contrast, the additive form can assign high scores when only one of these two conditions is met, leading to inflated anomaly scores in some dense regions. Nonetheless, we agree that complementary visualizations would make this distinction more tangible and interpretable. We will add visual examples showcasing the strengths of a multiplicative combination to Appendix Q in the revised version.
>
> ### Ensemble Scoring Mechanism
>
> We fully agree that demonstrating the robustness and variance reduction of the ensemble beyond theoretical bounds enhances the transparency and interpretability of our method. To address this, we will include a direct empirical comparison between the ensemble score and the top-performing single subset (Subset #1) across a diverse collection of datasets in the manuscript. The ensemble consistently yields higher AUC-ROC values than any individual subset, demonstrating its robustness and generalization benefits. The results confirm our theoretical claims and align with the intuition that averaging over multiple diverse scoring functions reduces sensitivity to idiosyncratic failure modes of individual subsets, thereby increasing robustness.
>
>
>
> **Suggestions**
>
> We appreciate the helpful suggestions of the reviewer that have already led to additional numerical results. In addition, new figures will be added to the final manuscript to illustrate specific components of our methodology.

---

> > ### Comment · Reviewer_TZiB · 2025-08-05
> >
> > Thank you for the authors' hard work. Regarding the last question, the authors stated: "To address this, we will include a direct empirical comparison between the ensemble score and the top-performing single subset (Subset #1) across a diverse collection of datasets in the manuscript." However, no such results are provided in the rebuttal. While visualization may be not available, the tabular results should still be presented to support this claim.

---

> > > ### Author Response · Authors · 2025-08-05
> > >
> > > We thank the reviewer for the valuable feedback.
> > >
> > >
> > > ## Empirical comparison of the ensemble vs. the best single subset
> > >
> > > The Table below presents the mean AUC‑ROC under the default protocol described in the main paper, comparing the proposed ensemble-based ADERH with the best-performing subset variant across various datasets.
> > >
> > >
> > >
> > > **Table: AUC-ROC performance comparison between the proposed ensemble-based ADERH and the best-performing subset.** Bold = best score (rank 1).
> > >
> > > | Dataset      | Ensemble ADERH | Subset (#1)    |
> > > |--------------|----------------|----------------|
> > > | Optdigits    | **0.775 (1)**   | 0.597 (2)     |
> > > | Wbc          | **1.000 (1)**   | 0.949 (2)     |
> > > | Lymphography | **1.000 (1)**   | 0.966 (2)    |
> > > | Celeba       | **0.732 (1)**   | 0.631 (2)    |
> > > | Skin         | **0.788 (1)**   | 0.577 (2)     |
> > > | Pendigits    | **0.962 (1)**   | 0.814 (2)     |
> > > | Wdbc         | **0.981 (1)**   | 0.915 (2)    |
> > > | Toothbrush   | **0.901 (1)**   | 0.802 (2)     |
> > > | Wpbc         | **0.554 (1)**   | 0.495 (2)     |
> > > | Leather      | **0.991 (1)**   | 0.900 (2)    |
> > > | Satimage-2   | **0.998 (1)**   | 0.974 (2)     |
> > > | Backdoor     | **0.889 (1)**   | 0.804 (2)     |
> > > | Stripe       | **0.986 (1)**   | 0.917 (2)     |
> > > | Shuttle      | **0.987 (1)**   | 0.824 (2)    |
> > > | Waveform     | **0.768 (1)**   | 0.629 (2)     |
> > > | Cardio       | **0.938 (1)**   | 0.820 (2)    |
> > > | Bottle       | **0.964 (1)**   | 0.881 (2)     |
> > > | Wine         | **0.839 (1)**   | 0.697 (2)     |
> > > | Musk         | **1.000 (1)**   | 0.855 (2)     |
> > >
> > > Across all datasets, the ensemble outperforms the best single subset, with a paired Wilcoxon signed-rank test with Bonferroni-Holm correction showing a statistically significant improvement ($\alpha = 0.05$, $p = 3.81 \times 10^{-6}$).
> > >
> > > The Table above has been incorporated into the revised manuscript. We will release the code post-review, per NeurIPS policy.
> > >
> > > We hope this resolves the remaining concern.

---

> > > > ### Comment · Reviewer_TZiB · 2025-08-06
> > > >
> > > > Thank you for your responses; most of my concerns have been addressed. I will adjust my rating accordingly. Additionally, I kindly suggest that the authors incorporate these responses and the visualizations they promised in the final version of the paper, as they would significantly enhance its clarity and overall rationale.

---

> > > > > ### Author Response · Authors · 2025-08-06
> > > > >
> > > > > We sincerely thank Reviewer TZiB for the thoughtful follow-up comment. We confirm that all additional experiments from our rebuttal have been fully incorporated into the revised manuscript. This includes the extended performance comparison between the ensemble-based ADERH and the best-performing single subset, as well as the inclusion of comparative results with UniCAD (ICML 2025). In addition, we now report statistical measures such as standard deviations across multiple runs to strengthen the empirical rigor of our evaluation.
> > > > >
> > > > > As discussed, we will also incorporate visualizations illustrating the individual and combined effects of Pitch and NDensity to provide clearer empirical support and improve interpretability. In the revised manuscript, we present four visualizations: (1) highlighting only Pitch values, (2) highlighting only NDensity values, (3) visualizing their additive combination, and (4) visualizing their multiplicative combination. Upon analyzing this set of plots, it becomes clear that only the multiplicative combination is able to accurately delineate the majority of the anomalous instances. This visual evidence directly supports our theoretical motivation for using a multiplicative scoring strategy.

---

### Note · Authors · 2025-08-13

Dear Area Chair and Reviewers,

We sincerely thank you for your time and constructive feedback. All comments have been addressed, and the manuscript now improves clarity, strengthens the evaluation, and broadens the evidence supporting our contributions. Below, we highlight the core improvements:

**Baselines & scope.** We expanded comparisons to recent SOTA methods (e.g., UniCAD, ICML ’25) and classical covering methods (GEM, DTM). These comparisons underscore **ADERH**’s novel design—multi-scale hyperspheres induced by random pairs with a multiplicative *Pitch* × *NDensity* score—which cleanly separates anomalies from normal data. In our experiments, **ADERH** wins on the majority of datasets with statistically significant gains (paired Wilcoxon with Holm correction, α = 0.05).

**Visual support.** To better support the method beyond the theoretical analysis, we added targeted visualizations: (1) Pitch-only, (2) NDensity-only, (3) additive combination, (4) multiplicative combination, plus histograms of hypersphere radii across multiple datasets to demonstrate the intended multi-scale coverage from random pairing.

**Reporting of statistical results.** We added standard deviations across the benchmark datasets, which demonstrate that **ADERH** exhibits the lowest standard deviation, suggesting it is the most stable and consistent method across various datasets.

**Extended parameter search.** To further support our findings—and per Reviewer nU8x’s suggestion—we ran a more comprehensive grid over 25 hyperparameter combinations for state-of-the-art anomaly approaches. Across this grid, **ADERH** consistently delivers strong and statistically significant gains over compared approaches (paired Wilcoxon with Holm correction, α = 0.05), while exhibiting a broad performance plateau that indicates robustness to hyperparameter choice.

**Minor fixes.** All minor issues—typos, wording, notation, and reference/citation formatting—have been corrected in the revision.



Sincerely,
*The Authors*

---

### Decision · Program_Chairs · 2025-09-17

**Decision:**

Accept (poster)

**Comment:**

The paper proposes a new anomaly detection algorithm. The idea behind the algorithm is that anomalous points are separated from normal points, while normal points cluster in small neighborhoods, anomalous points are far away from such local clusters. Based on this, the proposed algorithm draws hyperspheres around randomly chosen points to detect anomalous points. The approach is found to do well experimentally, and comes with theoretical analysis under certain separation assumption on the anomalies. The reviewers concerns were adequately address in the discussion phase. A suggestion to improve the paper is that it would be interesting to have more detailed experimental studies, including ablations to understand when and why the proposed method is better than previous techniques.